# Compositional Monte Carlo Tree Diffusion
# for Extendable Planning

**Jaesik Yoon**\*
KAIST & SAP
jaesik.yoon@kaist.ac.kr

**Hyeonseo Cho**
KAIST
hyeonseo.cho@kaist.ac.kr

**Sungjin Ahn**\*
KAIST & NYU
sungjin.ahn@kaist.ac.kr

## Abstract

Monte Carlo Tree Diffusion (MCTD) integrates diffusion models with structured tree search to enable effective trajectory exploration through stepwise reasoning. However, MCTD remains fundamentally limited by training trajectory lengths. While periodic replanning allows plan concatenation for longer plan generation, the planning process remains locally confined, as MCTD searches within individual trajectories without access to global context. We propose Compositional Monte Carlo Tree Diffusion (C-MCTD), a framework that elevates planning from individual trajectory optimization to reasoning over complete plan compositions. C-MCTD introduces three complementary components: (1) Online Composer, which performs globally-aware planning by searching across entire plan compositions; (2) Distributed Composer, which reduces search complexity through parallel exploration from multiple starting points; and (3) Preplan Composer, which accelerates inference by leveraging cached plan graphs.

## 1 Introduction

Diffusion models have emerged as a powerful framework for trajectory planning, particularly excelling in long-horizon tasks due to their ability to learn plans in a holistic manner [2, 15, 23]. This stands in contrast to conventional autoregressive approaches [3, 11, 12], which construct trajectories step-by-step. By generating trajectories as coherent wholes, diffusion planners can effectively address key challenges in long-term planning, such as sparse rewards and the accumulation of transition errors. However, these models often struggle to generate complex long-horizon plans rarely encountered in the training distribution.

To address this limitation, a growing body of work has explored inference-time scaling techniques, particularly by incorporating them into the diffusion denoising process [25, 27, 32, 33]. Among these approaches, Monte Carlo Tree Diffusion (MCTD) [32] represents a novel perspective by combining tree search with denoising. This integration enables MCTD to achieve state-of-the-art performance across a wide range of planning tasks.

MCTD combines causally scheduled denoising with Monte Carlo Tree Search (MCTS) [7], establishing a structured framework for deliberative planning that has shown strong performance across diverse planning domains. By decomposing trajectory generation into a subplan-level tree process, which decomposes a full plan into multiple subplans and systematically searches for the best combination of

---

\*Correspondence to Jaesik Yoon and Sungjin Ahn <jaesik.yoon@kaist.ac.kr and sungjin.ahn@kaist.ac.kr>.

39th Conference on Neural Information Processing Systems (NeurIPS 2025).

these subplans performing exploration and exploitation to escape local optima. It has demonstrated notable success in complex, long-horizon planning tasks [28] where conventional diffusion planners often struggle, representing a substantial advancement in generative planning methods.

Despite its strong performance, MCTD inherently struggles to generate plans that exceed the trajectory lengths observed during training. While existing periodic replanning strategies appear to address this length limitation, they suffer from a fundamental myopic decision-making problem that severely undermines their effectiveness [32]. Specifically, the replanning approaches make planning decisions based solely on local information within each limited planning horizon. It can lead to dead-end states or suboptimal paths due to the lack of consideration about the global path structure or long-term consequences. This fundamental limitation raises a key research question: *How can we enable MCTD to perform globally-aware planning that substantially exceeds training trajectory lengths while avoiding the myopic pitfalls of traditional replanning approaches?*

To address this question, we propose Compositional Monte Carlo Tree Diffusion (C-MCTD), an inference-time scaling framework designed to compose entire plans, enabling reasoning across plans rather than within subplans. The core instantiation of this framework, referred to as *Online Composer*, extends the tree search of MCTD to the plan composition through three key components. First, stitching-based tree extension connects individual diffusion-generated plans into a longer, coherent plan, enabling global reasoning beyond the limitations of isolated plan generation. Second, guidance sets as meta-actions provide configurable control parameters for the plan generation process. This mechanism enables the planner to generate targeted and adaptive high-quality plans, balancing exploration and exploitation according to its given guidance set. Third, fast replanning for simulation quickly approximates remaining trajectory segments using accelerated denoising methods, significantly reducing computational costs while preserving trajectory coherence during inference.

While Online Composer demonstrates strong performance across diverse environments, its sequential search procedure becomes inefficient in large state spaces due to the exponential growth in the number of candidate plan combinations. To address this challenge, we introduce two specialized variants: *Distributed Composer* and *Preplan Composer*. Distributed Composer leverages parallel processing and plan sharing across multiple search trees to mitigate the combinatorial explosion of the search space. Preplan Composer, in contrast, preconstructs a plan graph offline, enabling more efficient inference-time planning by reducing online search overhead and improving overall performance.

Experimental results demonstrate that the proposed C-MCTD methods significantly outperform standard replanning strategies based on MCTD across a variety of task settings. Notably, Preplan Composer achieves perfect success on the challenging pointmaze-giant task, which requires generating plans approximately 10× longer than the trajectories seen during training.

The main contributions of this paper are as follows. (1) We propose Compositional Monte Carlo Tree Diffusion (C-MCTD), a novel inference-time scaling framework that fundamentally addresses the myopic planning limitations of existing approaches by enabling tree search to compose plans. (2) We provide comprehensive validation demonstrating that C-MCTD significantly outperforms existing MCTD-based and alternative long-horizon planning approaches across diverse planning domains.

## 2 Preliminaries

**Terminology.**  This work involves multiple hierarchical concepts of trajectories that require clear distinction. A *plan* refers to a single diffusion-generated trajectory from the base planner. A *subplan* is a trajectory segment that corresponds to a node in MCTD's tree structure. A *stitched plan* concatenates multiple individual plans end-to-end to extend beyond the original planning horizon. Finally, a *full plan* represents a complete trajectory that successfully connects the start state to the goal state.

### 2.1 Diffusion models for planning

Diffusion models have been successfully applied to planning by reformulating it as a generative modeling problem [1, 2, 15, 23, 34], where trajectories are represented as sequential data comprising state-action pairs. Formally, a trajectory is denoted as $\mathbf{x} = [s_0; a_0, s_1; a_1, \ldots, s_T; a_T]$, where $T$ is the planning horizon and $(s_t, a_t)$ represents the state-action pair at time step $t$. During inference, the trajectory is iteratively refined from an initial random noise through a sequence of denoising steps, progressively converging toward a coherent plan.

This approach differs fundamentally from traditional planning algorithms like Probabilistic Road Maps (PRM) [16] and Rapidly-exploring Random Trees (RRT) [20]. Whereas traditional methods are online algorithms that actively explore the state space by querying an environment simulator or interacting with the physical world, diffusion planners operate in a fully offline setting. They learn to generate the trajectories from a static dataset of prior experiences, requiring no online environment access during the planning phase.

To ensure that generated trajectories align with specified task objectives, both classifier-free [14] and classifier-guided sampling [8] techniques have been explored in the context of diffusion planning [1, 2, 15]. In this work, we adopt the classifier-guided approach [15], which incorporates a guidance function $\mathcal{J}_\phi(\mathbf{x})$ to steer the generative process toward high-reward trajectories. The modified sampling distribution is defined as:

$$\tilde{p}_\theta(\mathbf{x}) \propto p_\theta(\mathbf{x}) \exp\left(\mathcal{J}_\phi(\mathbf{x})\right). \tag{1}$$

This guidance mechanism biases the diffusion model toward trajectories with higher expected returns, while retaining the flexibility of the original generative distribution.

Additionally, we adopt semi-autoregressive generation with causal noise schedule [2], which selectively denoises uncertain tokens (typically future timesteps) while preserving causal dependencies. This approach maintains temporal consistency and significantly improves long-horizon plan quality compared to standard diffusion generation.

## 2.2 Monte Carlo Tree Diffusion

Monte Carlo Tree Diffusion (MCTD) [32] reformulates trajectory planning as a tree search problem, structured around three key components.

**Denoising as tree rollout.** Unlike conventional MCTS approaches that perform rollouts at the individual state level, MCTD operates at the level of subplans—partitioned segments of the full trajectory. A trajectory is represented as $\mathbf{x} = [\mathbf{x}_1, \ldots, \mathbf{x}_S]$, where $\mathbf{x}$ denotes the complete trajectory and each $\mathbf{x}_s$ is a constituent subplan. MCTD leverages a semi-autoregressive generation strategy [2], resulting in the following factorization: $p(\mathbf{x}) \approx \prod_{s=1} p(\mathbf{x}_s|\mathbf{x}_{1:s-1})$. This formulation preserves the global coherence characteristic of diffusion models while enabling intermediate evaluations akin to MCTS rollouts, effectively reducing the depth of the search tree.

**Guidance levels as meta-actions.** To balance exploration and exploitation during subplan-level expansion, MCTD introduces guidance levels as meta-actions. For example, a subplan may be assigned a discrete control mode—such as GUIDE or NO_GUIDE—which determines whether it is sampled from the guided distribution (Equation 1) or the unconditional prior. This mechanism enables subplan-specific control over guidance, allowing for flexible and adaptive trade-offs between exploration and exploitation throughout the planning process.

**Jumpy denoising as fast simulation.** To enable efficient simulation, MCTD leverages fast denoising techniques such as Denoising Diffusion Implicit Models (DDIM) [30] to complete the remaining noisy subplans using fewer denoising steps. Given a partially generated trajectory $\mathbf{x}_{1:s}$, the remaining segments $\tilde{\mathbf{x}}_{s+1:S}$ are rapidly denoised as: $\tilde{\mathbf{x}}_{s+1:S} \sim p(\mathbf{x}_{s+1:S}|\mathbf{x}_{1:s}, g)$, where $g$ denotes the guidance level. This produces a complete trajectory $\tilde{\mathbf{x}}$ that can be used for evaluation, enabling fast and approximate rollout within the tree.

**Four operation steps of MCTD.** MCTD adapts the four canonical steps of Monte Carlo Tree Search (MCTS) to operate within the diffusion framework as follows:

**(1) Selection.** The tree is traversed from the root using a node selection policy such as UCT [19]. Unlike standard MCTS, each node in MCTD corresponds to an extended subplan rather than a single state, reducing tree depth and promoting higher-level abstraction. **(2) Expansion.** When a leaf node is reached, a meta-action—the guidance level $g$—is selected. A new subplan $\mathbf{x}_s$ is then sampled conditioned on $g$. **(3) Simulation.** After expansion, the remaining subplans are rapidly completed using accelerated denoising techniques such as DDIM [30]. Although approximate, this simulation step offers significant computational efficiency while retaining sufficient fidelity for reward-based evaluation using the reward function, $r(\mathbf{x})$. **(4) Backpropagation.** The resulting statistics, including visitation counts and trajectory rewards, are propagated upward through the tree to update value estimates and guide future search.

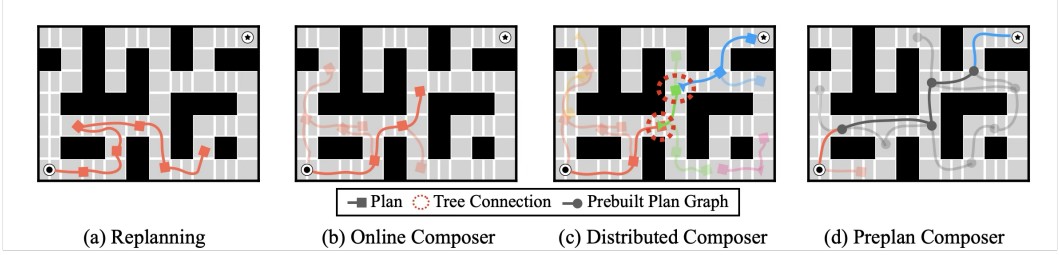

| ■ Plan | ⬭ Tree Connection | ● Prebuilt Plan Graph |

(a) Replanning      (b) Online Composer      (c) Distributed Composer      (d) Preplan Composer

Figure 1: **Comparison of stitched planning approaches**. **(a)** Replanning: Sequential plan generation without exploring alternatives, leading to myopic decisions. **(b)** Online Composer: Systematic tree search over plan combinations to avoid local optima. **(c)** Distributed Composer: Parallel tree growth from multiple origins with strategic connections (red circles) for efficient exploration. **(d)** Preplan Composer: Leverages prebuilt plan graphs for rapid solution composition with minimal online search.

## 3    Compositional Monte Carlo Tree Diffusion

While MCTD demonstrates strong performance on complex planning tasks, it is inherently limited in the training trajectory lengths. Existing replanning strategies (Figure 1 (a)) generate new plans when current plans end, but remain inherently myopic—making decisions based solely on local information, often leading to dead-end states or suboptimal paths.

To overcome this fundamental limitation, we propose *Compositional Monte Carlo Tree Diffusion (C-MCTD)*, a framework that constructs a globally coherent plan by composing shorter, high-quality plan segments within a tree search, effectively reasoning over long-horizon dependencies. We develop three variants targeting different scalability challenges:

- **Online Composer (OC)**: Systematic tree search for composing plans with coherent reasoning across them.
- **Distributed Composer (DC)**: Parallel tree expansion across multiple origins to reduce search depth.
- **Preplan Composer (PC)**: Graph-based planning using precomputed waypoint connections for computational efficiency.

### 3.1    Online Composer

We introduce **Online Composer (OC)**, the foundational variant of C-MCTD that addresses myopic planning through systematic plan-level tree search. OC integrates three key components: (1) stitching-based tree expansion that builds a search tree where each node represents an entire, generated plan, rather than partially denoised plan, (2) guidance sets as meta-actions for flexible inference-time scaling, and (3) fast replanning for efficient simulation. The approach is illustrated in Figure 1(b), with architecture details in Figure 2 and the core algorithm in Algorithm 1.

**Stitching-based tree expansion.** To implement plan-level tree search, OC performs stitching at each tree expansion by connecting the terminal state of the parent node's plan to the starting state of the newly generated plan. This enables each expansion to generate longer plans as the length of the plans generated from the diffusion planner [2, 15, 32] while systematically searching for global optima through tree exploration. The resulting trajectory is a stitched sequence of plans $\mathbf{x} = [\mathbf{x}^1, \ldots, \mathbf{x}^M]$, generated autoregressively:

$$p_\theta(\mathbf{x}) \approx \prod_{m=1}^{M} p_\theta(\mathbf{x}^m \mid \mathbf{x}^{1:m-1}), \tag{2}$$

where each $\mathbf{x}^m$ is a complete plan from the diffusion planner.

**Guidance sets as meta-actions.** In MCTD [32], the *guidance level*—a parameter controlling the influence of a guidance function on generation—is used as a meta-action for branching the search

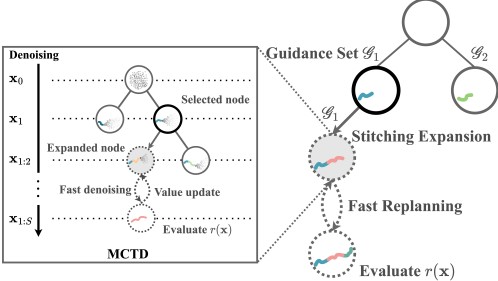

Figure 2: Architecture of Online Composer

**Algorithm 1** Online Composer

1: Initialize tree $\mathcal{T}_i$ on $s_{\text{start}}$
2: **while not** goal reached **do**
3:     Select node $v$ from $\mathcal{T}$
4:     Expand node $v$ via `Stitching`
5:     Simulate with `FastReplanning`
6:     Backpropagate rewards through $\mathcal{T}$
7: **end while**
8: **return** goal_reached_plan

tree. Our approach generalizes this concept to the plan level by introducing a guidance set as its meta-action. This set, comprising multiple guidance levels, enables inference-time planners (e.g., Best-of-$N$ or MCTD) to dynamically select the most appropriate level at each step. This mechanism empowers the planner to generate more effective and diverse local plans by adaptively modulating the guidance function's influence during the tree search. We empirically demonstrate the efficacy of using guidance sets in Appendix B.1. By enabling more sophisticated plan generation at each compositional step, this approach significantly enhances the quality of the final stitched solutions.

**Fast replanning for simulation.** To accelerate planning, OC employs fast replanning inspired by jumpy denoising [30] from MCTD. Once tree expansion reaches the $m$-th plan, the remaining trajectory is rapidly completed through a replanning process with the jumpy denoising that skips the denoising step every $C$ steps:

$$\tilde{\mathbf{x}}^{m+1:M} \sim p_\theta(\mathbf{x}^{m+1:M} \mid \mathbf{x}^{1:m}, \mathcal{G}), \tag{3}$$

producing a complete trajectory $\mathbf{x} = (\mathbf{x}^{1:m}, \tilde{\mathbf{x}}^{m+1:M})$ for direct evaluation via reward function $r(\mathbf{x})$. This approach reduces computational overhead while preserving long-horizon coherence. The effectiveness of this fast replanning is empirically analyzed in Appendix B.2.

**Scalability considerations.** While OC effectively extends planning beyond training trajectory lengths, it faces scalability challenges in extremely large spaces due to exponential search growth inherent in sequential tree search [7, 32]. These limitations are empirically demonstrated in our maze experiments (Section 5.2). To address these challenges, we introduce Distributed Composer and Preplan Composer in the following sections.

### 3.2 Distributed Composer

While Online Composer effectively explores plan combinations, the search space grows exponentially in long-horizon planning tasks, requiring prohibitively deep tree search. We address this challenge by introducing *Distributed Composer (DC)*, which parallelizes tree stitching across multiple starting positions as shown in Figure 1 (c). These starting positions are identified as cluster centroids from the training dataset, representing strategically significant states that are frequently traversed or serve as critical waypoints. Building on this framework, DC incorporates three key innovations: (1) guidance-oriented parallel tree search, (2) strategic tree connection, and (3) efficient path synthesis. The core algorithm of DC is discussed in Algorithm 2.

**Guidance-oriented parallel tree search.** A key challenge in parallel plan-level tree search is efficiently guiding expansion from multiple starting positions. While directly connecting all position pairs enables plan stitching, this requires quadratic computational overhead for the parallelism degree. Instead, DC optimizes search efficiency by guiding each tree's expansion toward task-relevant objectives. For each tree starting position $s_i \in \mathcal{S}$, the guided trajectory distribution is:

$$\tilde{p}_\theta(\mathbf{x}|s_i) \propto p_\theta(\mathbf{x}|s_i) \exp(\mathcal{J}_\phi(\mathbf{x})), \tag{4}$$

where $p_\theta(\mathbf{x}|s_i)$ is the base diffusion distribution from position $s_i$, and $\mathcal{J}_\phi(\mathbf{x})$ is a task-specific guidance function biasing exploration toward planning-relevant regions. This enables each tree to prioritize expansions likely to contribute to the overall task, with connections established via strategic tree connection.

| **Algorithm 2** Distributed Composer | **Algorithm 3** Preplan Composer |
|---|---|
| 1: Initialize trees $\mathcal{T}_i$ on origins $s_i \in S$ | 1: **Offline**: Build plan graph $G$ on $S$ |
| 2: Initialize connectivity graph $G = (\mathcal{S}, \emptyset)$ | 2: **for** $s_i \in S$ **in parallel do** |
| 3: **while not** synthesized plan found **do** | 3:     Apply Online Composer (Alg. 1) to find local paths from $s_{\text{start}}$ to $s_i$ |
| 4:     **for** $\mathcal{T}_i$ on $s_i \in S$ **in parallel do** | 4:     Apply Online Composer (Alg. 1) to find local paths from $s_i$ to $s_{\text{goal}}$ |
| 5:         Execute Four steps in Alg. 1 | 5: **end for** |
| 6:     **end for** | 6: Update $G$ with new connections |
| 7:     Update $G$ via strategic connections | 7: Apply shortest-path synthesis on $G$ |
| 8:     Apply shortest-path synthesis | 8: **return** synthesized_plan |
| 9: **end while** | |
| 10: **return** synthesized_plan | |

**Strategic tree connection.** Distributed Composer grows multiple search trees in parallel, each from a different starting position. A naive strategy would compare every node across all trees to identify connection opportunities, but this results in quadratic computational overhead. To avoid this, DC connects trees only when a plan from one tree reaches the starting position of another, narrowing the search to key waypoints and enabling efficient composition without exhaustive comparison. Specifically, for trees $\mathcal{T}_i$ and $\mathcal{T}_j$ with starting positions $s_i$ and $s_j$, a connection is attempted when:

$$\text{Connect}(\mathcal{T}_i, \mathcal{T}_j) = \begin{cases} \text{True}, & \text{if } \exists\, v \in \mathcal{T}_i : \min_{p \in v.\text{trajectory}} \text{dist}(p, s_j) < \epsilon \\ \text{False}, & \text{otherwise} \end{cases} \tag{5}$$

where $v$.trajectory is the position sequence of node $v$'s plan, and $\text{dist}(p, s_j)$ measures distance between position $p$ and starting position $s_j$. This strategically targets tree starting positions as natural environment waypoints. When a connection is established, the connectivity graph $G = (\mathcal{S}, E)$ is updated:

$$E = E \cup \{(s_i, s_j, v.\text{plan})\}, \tag{6}$$

where $v$.plan provides a feasible path from $s_i$ to $s_j$ and $\epsilon$ is a small distance threshold. This approach significantly reduces computational overhead by parallelizing search across multiple origins, avoiding exponential single-tree complexity. The distance threshold, $\epsilon$, is a key hyperparameter that can affect DC's performance. We empirically analyze this effect in Appendix B.3.

**Efficient path synthesis.** After parallel tree expansion and connection, DC constructs a connectivity graph $G$ where nodes are the starting positions $\mathcal{S}$ and edges represent feasible plans between them. DC then applies classical shortest-path algorithms (Dijkstra [9] or A* [13]) to find optimal plan combinations. Search terminates when the synthesized plan satisfies task constraints (e.g., maximum episode length), ensuring computational efficiency by avoiding unnecessary exploration once a viable solution is found.

**Computational challenges.** While DC reduces individual tree depth through parallelization, it still incurs substantial inference-time computation, particularly when exploring ultimately task-irrelevant positions. To address this inefficiency, we introduce Preplan Composer in the next section, which leverages precomputed knowledge for enhanced planning efficiency.

### 3.3 Preplan Composer

To address DC's computational challenges, we propose *Preplan Composer (PC)*, which amortizes planning costs through offline precomputation. As illustrated in Figure 1 (d), PC pre-builds a comprehensive **plan graph** offline, then leverages this graph during inference to guide tree search and dramatically reduce online computational overhead. The core algorithm is presented in Algorithm 3.

**Pre-building the plan graph.** Unlike DC's dynamic graph construction, PC builds a reusable graph by systematically exploring connections between selected starting positions. PC employs Online Composer to generate plans between waypoint pairs, guided by position-specific rather than task-specific guidance:

$$\tilde{p}_\theta(\mathbf{x}|s_i) \propto p_\theta(\mathbf{x}|s_i) \exp(-\text{dist}(\mathbf{x}, s_j)), \tag{7}$$

where $\text{dist}(\mathbf{x}, s_j)$ measures the minimum distance between any state in plan $\mathbf{x}$ and target position $s_j$. This *task-agnostic* process generates a reusable knowledge repository for multiple planning queries within the same environment. While this requires computationally intensive quadratic search between waypoints, the preprocessing phase allows substantial resource allocation to thoroughly sample the environment and establish reliable connections. This amortized computational overhead enables efficient online planning through the prebuilt graph, with each edge storing trajectory and cost information. A detailed analysis of this computational overhead is provided in Appendix B.5.

**Inference-time search.** When presented with a goal-conditioned planning task $(s_{\text{start}}, s_{\text{goal}})$, PC efficiently leverages the prebuilt plan graph for solution composition. The algorithm employs Online Composer to generate only short connecting plans: from $s_{\text{start}}$ to waypoints and from waypoints to $s_{\text{goal}}$. Since long-distance connections between waypoints are already planned in the preprocessing phase, OC focuses solely on these local connections. Using the augmented graph, PC applies shortest-path algorithms [9, 13] to identify the optimal waypoint sequence forming a complete solution. This approach dramatically reduces search complexity compared to DC's extensive online search, as most potential paths are pre-encoded in the graph.

## 4   Related work

**Diffusion models for planning.** Diffusion models [29] excel at long-horizon planning, particularly for sparse reward settings, by learning to generate plans holistically [1, 2, 15, 23, 34]. Research advances include hierarchical planning approaches for improved efficiency and long-term reasoning [5, 6, 10], hybrid planners integrating value learning policies [4], and diffusion planning with causal noise schedule for generating causally consistent plans [2].

**Inference-time scaling in diffusion planning.** Recent works have explored inference-time scaling to enhance diffusion planner reasoning capabilities [32, 33]. Yoon et al. [32] integrates semi-autoregressive denoising [2] with Monte Carlo Tree Search (MCTS) [7], while Zhang et al. [33] applies MCTS to diffusion models using learnable energy functions for value estimation.

**Planning beyond training trajectory length.** Several approaches extend planning capabilities beyond training trajectory lengths [6, 17, 21, 24], primarily by modifying the training data or objectives. For instance, some methods create longer training sequences by stitching trajectories from the dataset before training the diffusion planner [6, 21]. Others introduce specialized training objectives that encourage the model to learn compositional structures from sub-trajectories [24], or employ dedicated value functions for long-term estimation while generating short-horizon plans [17].

Our work uniquely applies inference-time scaling specifically for trajectory stitching, providing a training-free approach that extends planning capabilities without specialized training procedures or data augmentation.

## 5   Experiments

To demonstrate the effectiveness of C-MCTD in compositional long-horizon planning, we conduct a comprehensive evaluation on tasks from the Offline Goal-conditioned RL benchmark (OGBench) [28], following MCTD's experimental setup [32]. Our evaluation spans point and ant maze navigation with extended horizons, multi-cube robot arm manipulation, and partially observable visual maze tasks. All evaluated methods are diffusion planners, which, as discussed in Section 2.1, are trained on offline trajectory datasets and do not require environment interaction during training or planning. We report mean success rate (%) and planning time (seconds), averaged over 50 runs (5 tasks × 10 seeds), with complete configuration details in Appendix A.

### 5.1   Baselines

We compare C-MCTD against diverse baselines spanning stitching methods and diffusion planners. For stitching approaches, we evaluate **Replan** (fixed-interval plan regeneration without plan-level search) and **DatasetStitch** [6, 21] (training on concatenated trajectories). For diffusion planners, we include **Diffuser** [15] (foundational diffusion planning), **Diffusion Forcing** [2] (causal noise

Table 1: **Long-horizon maze navigation performance.** Success rates (%) for pointmaze and antmaze environments across medium, large, and giant maze scales. The models are trained from *stitch* datasets. Results are presented as mean ± standard deviation across multiple runs.

| Method | Success Rate (%) on PointMaze ↑ | | | Success Rate (%) on AntMaze ↑ | | |
|---|---|---|---|---|---|---|
| | medium | large | giant | medium | large | giant |
| Diffuser-Replan | $38 \pm 6$ | $40 \pm 0$ | $0 \pm 0$ | $46 \pm 18$ | $12 \pm 10$ | $0 \pm 0$ |
| Diffuser-DatasetStitch | $32 \pm 22$ | $0 \pm 0$ | $0 \pm 0$ | $12 \pm 10$ | $0 \pm 0$ | $0 \pm 0$ |
| Diffusion Forcing (DF) | $53 \pm 16$ | $20 \pm 0$ | $0 \pm 0$ | $60 \pm 13$ | $26 \pm 9$ | $0 \pm 0$ |
| DF-DatasetStitch | $52 \pm 23$ | $26 \pm 13$ | $0 \pm 0$ | $24 \pm 8$ | $6 \pm 9$ | $0 \pm 0$ |
| SSD | $40 \pm 0$ | $40 \pm 0$ | $0 \pm 0$ | $12 \pm 9$ | $0 \pm 0$ | $0 \pm 0$ |
| CompDiffuser | $\mathbf{100} \pm 0$ | $\mathbf{100} \pm 0$ | $68 \pm 3$ | $96 \pm 2$ | $86 \pm 2$ | $65 \pm 3$ |
| MCTD-Replan | $90 \pm 14$ | $20 \pm 0$ | $0 \pm 0$ | $75 \pm 8$ | $30 \pm 10$ | $0 \pm 0$ |
| MCTD-DatasetStitch | $12 \pm 10$ | $2 \pm 6$ | $0 \pm 0$ | $87 \pm 13$ | $6 \pm 9$ | $0 \pm 0$ |
| Online Composer | $93 \pm 9$ | $82 \pm 17$ | $0 \pm 0$ | $\mathbf{98} \pm 6$ | $\mathbf{94} \pm 9$ | $12 \pm 16$ |
| Distributed Composer | $\mathbf{100} \pm 0$ | $\mathbf{100} \pm 0$ | $26 \pm 21$ | $\mathbf{98} \pm 6$ | $93 \pm 9$ | $21 \pm 14$ |
| Preplan Composer | $\mathbf{100} \pm 0$ | $\mathbf{100} \pm 0$ | $\mathbf{100} \pm 0$ | $94 \pm 9$ | $\mathbf{94} \pm 9$ | $\mathbf{75} \pm 18$ |

Table 2: **Run time and plan quality comparison.** The run time (sec.) on tree search and the generated plan quality (the required interaction steps to reach the goal) on the pointmaze environment with medium, large, and giant mazes when trained from the *stitch* datasets.

| Method | Run Time (sec.) ↓ | | | Plan Length ↓ | | |
|---|---|---|---|---|---|---|
| | medium | large | giant | medium | large | giant |
| Online Composer | $83.6 \pm 31.8$ | $91.2 \pm 35.7$ | $269.8 \pm 51.0$ | $595.7 \pm 56.1$ | $743.5 \pm 53.6$ | $1000.0 \pm 0.0$ |
| Distributed Composer | $26.7 \pm 5.8$ | $39.1 \pm 5.0$ | $530.7 \pm 108.8$ | $493.3 \pm 98.2$ | $608.5 \pm 75.1$ | $974.7 \pm 24.6$ |
| Preplan Composer | $\mathbf{11.1} \pm 0.2$ | $\mathbf{21.5} \pm 0.4$ | $\mathbf{36.9} \pm 0.3$ | $\mathbf{143.8} \pm 1.1$ | $\mathbf{256.8} \pm 0.9$ | $\mathbf{451.9} \pm 1.9$ |

scheduling for long-horizon control), **SSD** [17] (stitching diffusion planner via learned value function guidance), **CompDiffuser** [24] (stitching diffusion planner with sub-trajectory dependency modeling), and **MCTD** [32] (direct comparison for plan-level vs. in-plan search). Complete details are provided in Appendix A.3.

## 5.2 Long-horizon maze navigation: scaling beyond trained trajectory length

We evaluate the effectiveness of C-MCTD on long-horizon tasks using PointMaze and AntMaze environments from OGBench, which require planning trajectories up to 1000 steps despite training on significantly shorter sequences (100 steps). Following prior work [15, 32], PointMaze uses a heuristic controller while AntMaze employs a learned value-based policy [31].

**Baseline performance degrades with complexity**. Table 1 reveals that conventional stitching methods (Replan, DatasetStitch) exhibit severe performance drops as maze complexity increases, with most methods achieving 0% success on giant mazes. Notably, CompDiffuser achieves strong performance on smaller mazes but degrades sharply in giant environments (68% vs. 100% on medium/large), highlighting the critical need for flexible plan-level search to escape the local optima in complex long-horizon scenarios.

**C-MCTD demonstrates superior scalability**. Our methods show strong robustness across all maze sizes. **Preplan Composer achieves perfect 100% success on PointMaze-Giant**, significantly outperforming the best baseline (CompDiffuser at 68%). Online Composer excels on medium and large mazes but struggles in giant environments due to exponential search space growth. Distributed Composer outperforms Online Composer on medium and large mazes through parallel tree stitching. However, it faces challenges in giant mazes, as its synthesized plans often fail to meet the strict episode length constraint. When we relaxed the maximum episode length from 1000 to 2000 steps, DC's success rate improved to 86%. The qualitative result comparison between C-MCTD variants is shown in Figure 4.

**Efficiency analysis reveals computational trade-offs**. Table 2 demonstrates that Distributed and Preplan Composers achieve superior efficiency through parallelism and amortized graph structures

Table 3: **Robot arm cube manipulation performance.** Success rates (%) across increasing complexity manipulation tasks involving single, double, triple, and quadruple cube arrangements in OGBench, presented as mean ± standard deviation.

| Method | Success Rate (%) ↑ | | | |
|---|---|---|---|---|
| | **single** | **double** | **triple** | **quadruple** |
| Diffuser-Replan | $92 \pm 13$ | $12 \pm 13$ | $4 \pm 8$ | $0 \pm 0$ |
| Diffusion Forcing | $100 \pm 0$ | $18 \pm 11$ | $16 \pm 8$ | $0 \pm 0$ |
| MCTD-Replan | $100 \pm 0$ | $78 \pm 11$ | $40 \pm 21$ | $24 \pm 8$ |
| Online Composer with Plan Cache | $\mathbf{100} \pm 0$ | $\mathbf{100} \pm 0$ | $\mathbf{75} \pm 12$ | $\mathbf{82} \pm 11$ |

respectively. Preplan Composer produces notably shorter, higher-quality plans through its pre-built graph approach, achieving the best computational efficiency while maintaining perfect success rates.

## 5.3 Multi-object manipulation: compositional planning with plan caching

We evaluate C-MCTD on multi-cube manipulation tasks from OGBench [28], which require precise action sequencing for stacking cubes into predefined configurations. These tasks demand compositional reasoning to determine optimal cube movement sequences, making them ideal for testing plan-level search capabilities. Our approach combines diffusion-based high-level planning with low-level value-guided control and object-wise guidance as described in prior work [4, 32].

**Method adaptation for manipulation tasks**. For these specialized tasks, we exclude Distributed and Preplan Composers since parallel tree stitching from random positions proves inefficient for precise manipulation sequences. Instead, we implement a *plan cache mechanism* that stores and retrieves previously generated plans for identical scenarios, significantly enhancing search efficiency (detailed analysis in Appendix B.4).

**Baseline performance degrades with task complexity**. Table 3 shows that while MCTD-Replan outperforms Diffuser-based approaches due to object-wise guidance capabilities [32], its effectiveness diminishes substantially as complexity increases—dropping from 100% (single cube) to 24% (quadruple cube). This performance degradation highlights the limitations of the baseline's search strategy in complex sequential manipulation tasks.

**Online Composer excels through compositional search**. In contrast, our Online Composer with Plan Cache demonstrates robust performance across all complexity levels. **It achieves a 100% success rate on single and double cube tasks and maintains strong performance on more difficult scenarios, succeeding in 75% of triple-cube and 82% of quadruple-cube tasks.** These results underscore the effectiveness of our compositional search approach, where the plan cache enables the efficient reuse of sub-solutions across related manipulation sequences.

## 5.4 High-dimensional visual navigation: POMDP planning challenges

We evaluate C-MCTD on Visual Maze tasks from prior work [32], which require navigation from starting observations to target goals within $64 \times 64$ RGB observation spaces—forming challenging Partially Observable Markov Decision Processes (POMDPs). Following established protocols [32], we employ Variational Autoencoders [18] for latent space planning, inverse dynamics models for action generation, and positional estimators for POMDP guidance (details in Appendix A.5.3).

Table 4: **Visual PointMaze results.**

| Method | Success Rate (%) ↑ | |
|---|---|---|
| | **medium** | **large** |
| Diffuser | $8 \pm 13$ | $0 \pm 0$ |
| Diffuser-Replan | $8 \pm 10$ | $0 \pm 0$ |
| Diffusion Forcing | $66 \pm 32$ | $8 \pm 12$ |
| MCTD | $82 \pm 18$ | $0 \pm 0$ |
| MCTD-Replan | $90 \pm 9$ | $20 \pm 21$ |
| Online Composer | $\mathbf{100} \pm 0$ | $\mathbf{54} \pm 18$ |
| Distributed Composer | $94 \pm 6$ | $26 \pm 9$ |
| Preplan Composer | $96 \pm 8$ | $48 \pm 16$ |

**Myopic planning fails in complex visual environments**. Table 4 shows that while MCTD-Replan achieves strong performance on medium mazes (90%), it suffers substantial degradation on large mazes (20%), demonstrating the limitations of myopic plan generation in high-dimensional partially observable settings.

**Online Composer excels through plan-level reasoning**. **Online Composer significantly outperforms all baselines**, achieving perfect success on medium mazes and maintaining strong performance

on large mazes (54% vs. 20% for MCTD-Replan). This demonstrates the effectiveness of plan-level tree search for overcoming myopic limitations in visual navigation tasks.

**High-dimensional challenges limit parallel variants**. Interestingly, Distributed and Preplan Composers underperform compared to Online Composer—contrasting with maze experiments (Section 5.2). Our analysis reveals that clustering meaningful waypoints in high-dimensional latent spaces proves challenging, hindering efficient plan integration. This limitation highlights an important direction for future research in visual POMDP planning.

## 6 Discussion

In this work, we introduced Compositional Monte Carlo Tree Diffusion (C-MCTD), a novel framework that integrates plan-level search into diffusion-based planners. This approach enables compositional reasoning, allowing for generalization to long-horizon tasks far exceeding the scope of the training data. While our experiments demonstrate significant performance gains, it is important to discuss the inherent trade-offs of our method and the broader challenges that remain for diffusion planning models.

**Limitations and trade-offs of C-MCTD variants.** The primary challenge of C-MCTD is the computational cost associated with searching a vast compositional plan space. We introduced Distributed Composer (DC) and Preplan Composer (PC) to mitigate this cost through parallelism and amortized graph search, respectively. However, these variants introduce their own trade-offs. PC requires a pre-computation to build the graph. It is highly efficient and is suited for the environments where the graph can be reused across many queries. DC's parallel tree stitching is powerful for spatial navigation but can be less efficient for tasks requiring precise sequential dependencies, as seen in our manipulation experiments in Section 5.3. These trade-offs highlight that the optimal C-MCTD variant is task-dependent, a key insight for future applications.

**Future directions for diffusion planning models.** Looking forward, a key challenge for C-MCTD and other diffusion planners is achieving real-time inference speeds suitable for online robotics applications. Bridging this efficiency gap is a critical avenue for future work. Furthermore, our work inherits two fundamental limitations of the current generative planning paradigm. First, generalization to entirely novel environments remains difficult, as planners can produce kinematically invalid plans when faced with out-of-distribution states. Second, handling stochastic dynamics is an open problem, as standard diffusion models may generate an ineffective "averaged" plan rather than a robust, multi-modal policy.

Despite these open challenges, C-MCTD marks a critical step toward creating more general, scalable, and compositional generative planners, paving the way for solving increasingly complex decision-making problems.

## 7 Conclusion

We introduced Compositional Monte Carlo Tree Diffusion (C-MCTD), a novel framework that scales diffusion-based planners to generate complex, long-horizon plans by composing shorter trajectories at inference time. C-MCTD integrates three complementary approaches: the **Online Composer** for flexible, on-the-fly plan generation; the **Distributed Composer** for scaling through parallelism; and the **Preplan Composer** for maximum efficiency via pre-computed plan graphs. Our extensive evaluations show that this compositional framework significantly outperforms conventional replanning strategies. Notably, the Preplan Composer solves tasks requiring trajectories up to $10\times$ longer than those seen during training. These results demonstrate that compositional scaling at inference time can dramatically enhance the reasoning capabilities of diffusion planners. Crucially, this is achieved without any model retraining, offering a practical and effective path toward more capable agents for challenging sequential decision-making problems.

## Acknowledgments and Disclosure of Funding

We would like to extend our gratitude to Doojin Baek for his insightful discussions and assistance throughout this project. This research was supported by Brain Pool Plus Program (No.

2021H1D3A2A03103645) through the National Research Foundation of Korea (NRF) funded by the Ministry of Science and ICT (MSIT), Institute of Information & communications Technology Planning & Evaluation (IITP) grant funded by the Korea government (MSIT) (No. RS-2024-00509279, Global AI Frontier Lab), and GRDC (Global Research Development Center) Cooperative Hub Program (RS-2024-00436165) through the National Research Foundation of Korea (NRF) funded by the Ministry of Science and ICT (MSIT).

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

| (a) | (b) | (c) |

Figure 3: **Illustrations of the environments.** Our method is evaluated on three distinct tasks: (a) A long-horizon maze where the agent must navigate from a start to a goal state. (b) A robotic manipulation task requiring precise control of a robotic arm. (c) A visual maze task where observations are provided as first-person view images.

# A    Experiment details

## A.1    Computation resources

All experiments were conducted on high-performance hardware consisting of 8 NVIDIA RTX 4090 GPUs, 512GB system memory, and a 96-thread CPU. Training each model required about 6 hours, while inference for comprehensive experimental evaluation took up to 1 hour in the most computationally intensive cases.

## A.2    Environment details

To evaluate our C-MCTD, we adopt the benchmarks from Yoon et al. [32]. While the robotic manipulation and visual maze tasks replicate the environmental settings of the prior work, we introduce a more demanding evaluation for the long-horizon maze environments. Specifically, we employ the *stitch* datasets from Park et al. [28] rather than the *navigate* datasets. This choice creates a challenging scenario that requires generating plans significantly longer than the trajectories observed during training. For instance, training trajectories are limited to 200 steps, whereas successfully reaching goals in the test environments requires plans of up to 1000 steps. This setting allows us to rigorously assess the extendable planning capabilities of our approach. An overview of each task is provided in Figure 3.

## A.3    Baselines

To evaluate C-MCTD, we compare its performance against diverse baselines. These baselines combine different stitching methods with various diffusion planners to extend their effective planning horizons. We describe the core components below.

### A.3.1    Stitching methods

**Replan.**    This baseline addresses the fixed-horizon limitation of standard diffusion planners, which are constrained by the trajectory lengths in the training data. The **Replan** method operates by iteratively generating a new plan from the final state of the previous one. While this enables extendable planning, the myopic nature of each planning step often results in suboptimal long-horizon solutions.

**DataStitch.**    [6, 21] An alternative approach, which we term **DataStitch**, extends the planning horizon by first elongating the trajectories within the training dataset itself. Because trajectories in the original dataset are not necessarily connectable, this method involves training auxiliary diffusion planners and inverse dynamics models to generate synthetic continuations, effectively stitching existing data points together. For instance, Chen et al. [6] iteratively applied this process to extend trajectories up to sevenfold. Following this methodology, we create an extended version of the *stitch* dataset by lengthening trajectories to five times their original length and subsequently train our models on this augmented data.

### A.3.2 Diffusion planners

**Diffuser.** [15] As a primary baseline, we include Diffuser, which pioneered training diffusion models on sequences of concatenated state-action pairs. We evaluate Diffuser with both the Replan and DataStitch extension methods by training it on the corresponding augmented datasets.

**Diffusion Forcing.** [2] This Diffuser variant tokenizes data and applies varied noise levels to enable causal denoising that prioritizes near-future states. We employ the Transformer-based implementation proposed by Chen et al. [2]. As the method is inherently designed for replanning, we evaluate it with the DataStitch extension but do not create a separate "Replan" variant.

**Sub-trajectory Stitching with Diffusion (SSD).** [17] Kim et al. [17] introduced a diffusion planner that achieves extendability via a learned value function, which guides the model to concatenate short trajectories into a longer, task-relevant plan. While effective, we empirically found that learning a reliable value function is challenging with suboptimal data, whereas inference-time scaling offers a more robust alternative. Because SSD is natively designed for extendable planning, we do not evaluate it using our Replan or DataStitch extensions.

**Compositional Diffuser (CompDiffuser).** [24] Luo et al. [24] proposed an alternative approach to extendable planning where a diffusion model is trained to generate sub-trajectories conditioned on nearby context. This framework allows the model to compose multiple sub-trajectories, generating plans that are longer than any seen during training. Similar to SSD, CompDiffuser's native extendability makes evaluation with our proposed extensions unnecessary.

**Monte Carlo Tree Diffusion (MCTD).** [32] Yoon et al. [32] introduced a framework that implements inference-time scaling by branching the denoising process into a Monte Carlo Tree Search, rather than merely increasing sequential denoising steps. Guided by a reward function, this approach explores alternative paths to find optimal solutions, demonstrating superior performance across diverse planning tasks. We evaluate MCTD with both the Replan and DataStitch extension methods.

### A.4 Model hyperparameters

Our hyperparameter configuration is largely adopted from Yoon et al. [32]. For full reproducibility, we provide the detailed configurations for the baseline models in Tables 5–8. Hyperparameters introduced by our proposed C-MCTD are detailed in Section A.5 alongside their corresponding experimental setups. For the SSD baseline, we utilized the default configurations from its official public implementation[2].

### A.5 Evaluation details

This section outlines the configurations for each benchmark.

### A.5.1 Long-horizon maze environments

**Datasets and environment configuration.** To evaluate extendable planning, all models were trained on the challenging *stitch* datasets. An exception was made for the value-learning policy [31], which failed to converge on this data; as this baseline is not our focus, we trained it on the original *navigate* dataset. For all maze environments, we set the training trajectory length and the single-pass planning horizon to 100, sampling sub-sequences from the full 200-step trajectories to enhance data diversity. Following prior work [32], we remove random noise from the start and goal positions to isolate the planner's performance from environmental stochasticity. Furthermore, we employ a heuristic controller for PointMaze environments [15] and a value-based policy for AntMaze environments [4, 31], for which we set the subgoal interval to 10.

**Guidance mechanism.** To steer the diffusion process, we apply a guidance function that minimizes the L2 distance between each planned state $x_i$ and the goal $g$ [2, 32]. This function, defined as $\sum_i ||x_i - g||_2$, is used across all models. The strength of this guidance is controlled by a scale hyperparameter, which we configure for each baseline as follows:

---

[2] https://github.com/rlatjddbs/SSD

Table 5: Hyperparameters for the Diffuser baseline. These settings are adopted from the original MCTD paper [32] to ensure a fair comparison. Task-specific parameters, such as the planning horizon, are detailed in their respective sections.

| Hyperparameter | Value |
|---|---|
| *Training & Optimizer Settings* | |
| Learning Rate | $2 \times 10^{-4}$ |
| Batch Size | 32 |
| Max Training Steps | 20,000 |
| EMA Decay | 0.995 |
| Floating-Point Precision | 32-bit (FP32) |
| *Diffusion & Guidance Settings* | |
| Beta Schedule | Cosine |
| Objective | $x_0$-prediction |
| Guidance Scale | 0.1 |
| Open-loop Horizon | 50 (for Diffuser-Replan) |
| *Model Architecture (U-Net)* | |
| Depth | 4 |
| Kernel Size | 5 |
| Channel Sequence | $32, 128, 256$ |

- **Diffusion Forcing:** Following Chen et al. [2], we set the guidance scale to 3 for medium maps and 2 for large and giant maps.

- **MCTD:** The guidance sets are adopted from Yoon et al. [32]: $\{0, 0.1, 0.5, 1, 2\}$ for Point-Maze (Medium & Large), $\{0.5, 1, 2, 3, 4\}$ for PointMaze (Giant), and $\{0, 1, 2, 3, 4, 5\}$ for all AntMaze tasks.

Our proposed C-MCTD variants inherit these settings; during a node expansion, each of the two child nodes is assigned the corresponding guidance set for the environment.

**Reward function for tree search-based planners.** Both MCTD and C-MCTD require a reward function to evaluate simulated trajectories. We adopt the function from Yoon et al. [32], which incorporates two components. First, it penalizes physically implausible plans by assigning a reward of 0 to any trajectory with unrealistic position changes between consecutive states. Second, for valid trajectories that first reach the goal at timestep $t$, the reward is calculated as $r = (H - t)/H$, where $H$ is the maximum horizon length. This encourages finding shorter, more efficient paths.

**C-MCTD configurations.**

- **Node expansion planner:** For node expansion in the maze environments, C-MCTD utilizes a best-of-$N$ search-based planner [34] built upon Diffusion Forcing [2]. In this approach, $N$ candidate trajectories are generated, and the one yielding the highest reward (as defined above) is selected. We set $N = 50$ for all maze experiments.

- **DC and PC variants:** Our Distributed Composer (DC) and Preplan Composer (PC) variants identify representative waypoints via k-means clustering [22, 26] on the training data, with 10, 30, and 70 cluster centers for medium, large, and giant maps, respectively. For DC, the search is capped at 100 iterations. For PC, we limit the search to 20 iterations and 2 plan concatenations (a tree depth of 2) for medium and large maps; these are reduced to 10 iterations and 1 concatenation for the giant map. These settings render our methods substantially more efficient than the Online Composer baseline, which requires up to 500 search iterations and 10 concatenations.

Table 6: Hyperparameters for the Diffusion Forcing baseline. Settings are based on the implementation in Yoon et al. [32] and adapted for our tasks. Parameters that vary across environments, such as the guidance scale, are specified in Section A.5.

| Hyperparameter | Value |
|---|---|
| *Training & Optimizer Settings* | |
| Learning Rate | $5 \times 10^{-4}$ |
| Weight Decay | $1 \times 10^{-4}$ |
| Warmup Steps | 10,000 |
| Batch Size | 1024 |
| Max Training Steps | 200,005 |
| Training Precision | 16-bit (Mixed) |
| Inference Precision | 32-bit (FP32) |
| *Diffusion & Sampling Settings* | |
| Beta Schedule | Linear |
| Objective | $x_0$-prediction |
| Scheduling Matrix | Pyramid |
| Stabilization Level | 10 |
| DDIM Sampling $\eta$ | 0.0 |
| Frame Stack | 10 |
| Open-loop Horizon | 50 |
| Causal Mask | Not Used |
| *Model Architecture (Transformer)* | |
| Embedding Dimension | 128 |
| Number of Layers | 12 |
| Number of Attention Heads | 4 |
| FFN Dimension | 512 |

### A.5.2 Robot arm manipulation environment

**Environment and baseline configuration.** We conduct these experiments using the *play* datasets. The planning horizon is set to 200 for single-cube tasks, corresponding to the maximum episode length, and extended to 500 for more complex multi-cube tasks. Similar to the AntMaze experiments, we employ a value-based policy [31] as a low-level controller, with a subgoal interval of 10.

**Guidance mechanism.** The goal-reaching guidance function is identical to that used in the maze environments (Section A.5.1). For the Diffusion Forcing baseline, the guidance scale is set to 2. For tree-search planners (MCTD and C-MCTD), we adopt the object-wise guidance strategy from Yoon et al. [32]. This approach guides the planner to manipulate a single cube at a time, preventing physically infeasible plans that attempt to move multiple objects concurrently. The guidance set for this strategy is $\{1, 2, 4\}$, which our C-MCTD variants inherit for two child nodes during expansion.

**Reward function.** We adopt the reward function from Yoon et al. [32], which is designed to filter out physically unrealistic trajectories. A plan receives a reward of 0 if it violates any of the following conditions: (1) moving multiple objects simultaneously; (2) leaving an object in an unstable, mid-air position; (3) causing inter-object collisions; or (4) attempting to grasp an object obstructed by another. Otherwise, a positive reward is assigned based on task success.

**Macro-actions for planning efficiency.** Following Yoon et al. [32], we incorporate a set of pre-defined macro-actions (scripted primitives) to improve the efficiency and robustness of object-wise planning. These routines handle common operations, such as automatically releasing an object at its target or pre-positioning the arm before manipulating the next object. This scripting approach prevents common planning failures, like generating trajectories beyond the arm's kinematic reach or attempting to grasp with an occupied gripper.

Table 7: Hyperparameters for the Monte Carlo Tree Diffusion (MCTD) baseline. These settings are based on the original implementation [32]. Task-dependent parameters, such as the planning horizon and guidance set, are specified in Section A.5.

| Hyperparameter | Value |
|---|---|
| *Training & Optimizer Settings* | |
| Learning Rate | $5 \times 10^{-4}$ |
| Weight Decay | $1 \times 10^{-4}$ |
| Warmup Steps | 10,000 |
| Batch Size | 1024 |
| Max Training Steps | 200,005 |
| Training Precision | 16-bit (Mixed) |
| Inference Precision | 32-bit (FP32) |
| *Diffusion & Sampling Settings* | |
| Beta Schedule | Linear |
| Objective | $x_0$-prediction |
| DDIM Sampling $\eta$ | 0.0 |
| Frame Stack | 10 |
| Open-loop Horizon (MCTD-Replan) | 50 |
| Causal Mask | Not Used |
| Scheduling Matrix | Pyramid |
| Stabilization Level | 10 |
| *MCTD Search Settings* | |
| Max Search Iterations | 500 |
| Partial Denoising Steps | 20 |
| Jumpy Denoising Interval | 10 |
| *Model Architecture (Transformer)* | |
| Embedding Dimension | 128 |
| Number of Layers | 12 |
| Number of Attention Heads | 4 |
| FFN Dimension | 512 |

**C-MCTD configurations.** For the robot manipulation tasks, our C-MCTD employs two specific strategies:

- **Node expansion planner:** Unlike in the maze environments, node expansion utilizes the full MCTD planner instead of the simpler best-of-$N$ search. This choice is necessitated by the significantly longer planning horizons in this domain (200–500 steps), which demand a more powerful and systematic search capability.

- **Plan caching:** To improve computational efficiency, we introduce a *plan caching* strategy. Successfully generated plans are stored with their metadata (target object, start, and goal states). A cached plan is reused for a new task if its target object is identical and its start/goal states fall within an L2 distance of $\epsilon$ from the cached ones.

### A.5.3 Visual maze environment

**Environment and data configuration.** We follow the experimental setup from Yoon et al. [32] for the visual maze tasks. Using the data generation scripts from Park et al. [28], we create datasets of 1000-step trajectories for both medium and large maps. To ensure diverse training samples, our diffusion models are trained on sub-trajectories with a planning horizon of 500, sampled from these full-length trajectories.

Table 8: Hyperparameters for the value-learning policy baseline. These settings are configured for training the policy on the *navigate* dataset.

| Hyperparameter | Value |
| --- | --- |
| *Training & Optimizer Settings* | |
| Learning Rate | $3 \times 10^{-4}$ |
| Training Epochs | 2,000 |
| Gradient Clipping Norm | 7.0 |
| *Algorithm-Specific Settings* | |
| Learning $\eta$ | 1.0 |
| Max Q Backup | False |
| Reward Tune | cql_antmaze |
| Top-$k$ | 1 |
| Target Update Steps | 10 |
| Data Sampling Randomness ($p$) | 0.2 |

**Vision-based modeling pipeline.** To handle the high-dimensional visual inputs and partial observability, we adopt the modeling pipeline from Yoon et al. [32], which consists of three pre-trained components that operate on $64 \times 64 \times 3$ RGB observations:

- **Variational Autoencoder (VAE):** A VAE [18] is first pre-trained to encode each image observation into a compact 8-dimensional latent representation, $z_t$. This latent space serves as the basis for all subsequent planning and control models.

- **Inverse dynamics model:** An MLP-based inverse dynamics model, $f_{\text{inv}}$, is trained to predict the action $\hat{a}_t$ required to transition between latent states. To infer velocity from static images, the model is conditioned on three consecutive latent states: $\hat{a}_t = f_{\text{inv}}(z_{t-1}, z_t, z_{t+1})$. This learned model acts as a low-level controller.

- **Position estimator:** A separate MLP is trained to predict the agent's approximate $(x, y)$ coordinates from a given latent state $z_t$. This provides a weak positional signal for planning guidance without compromising the task's partial observability.

**Guidance Mechanism.** Guidance is performed in the state space using the positional signal from the pre-trained position estimator. This allows us to apply the same L2 distance-based guidance function as described in Appendix A.5.1. Following Yoon et al. [32], we use a guidance set of $\{0, 0.1, 0.5, 1, 2\}$ for all baselines. Our C-MCTD variants inherit this configuration, assigning this guidance set to each child node during an expansion.

**C-MCTD Configurations.** For our Distributed Composer (DC) and Preplan Composer (PC) variants, we generate representative waypoints by applying k-means clustering to the coordinates predicted by the position estimator. We define 3 and 5 cluster centers for the medium and large maps, respectively.

Table 9: **The guidance sets meta-action ablation study on PointMaze environments.** Success rate (%) comparison with the variant of Online Composer with fixed guidance level as a meta-action rather than guidance set, presented as mean $\pm$ standard deviation.

| Environment | Success Rate (%) $\uparrow$ | | | | |
|---|---|---|---|---|---|
| | **Default** | **Fixed: 0.1** | **Fixed: 0.5** | **Fixed: 1** | **Fixed: 2** |
| PointMaze Medium | $\mathbf{93}_{\pm 9}$ | $80_{\pm 0}$ | $90_{\pm 10}$ | $\mathbf{98}_{\pm 6}$ | $83_{\pm 8}$ |
| PointMaze Large | $\mathbf{82}_{\pm 17}$ | $77_{\pm 14}$ | $52_{\pm 13}$ | $27_{\pm 10}$ | $20_{\pm 0}$ |

Table 10: **The Fast replanning ablation study on PointMaze environments.** Success rate (%) comparison with the variant of Online Composer without fast replanning, presented as mean $\pm$ standard deviation. The numbers in parentheses indicate the maximum search iterations allowed.

| Enivornment | Success Rate (%) $\uparrow$ | | | | | |
|---|---|---|---|---|---|---|
| | **W/ FR (200)** | **W/o FR (200)** | **W/ FR (100)** | **W/o FR (100)** | **W/ FR (50)** | **W/o FR (50)** |
| PointMaze Medium | $\mathbf{96}_{\pm 8}$ | $82_{\pm 6}$ | $\mathbf{94}_{\pm 9}$ | $76_{\pm 12}$ | $\mathbf{92}_{\pm 10}$ | $63_{\pm 15}$ |
| PointMaze Large | $\mathbf{84}_{\pm 15}$ | $54_{\pm 13}$ | $\mathbf{82}_{\pm 14}$ | $49_{\pm 10}$ | $\mathbf{82}_{\pm 19}$ | $32_{\pm 11}$ |

# B  Additional experimental results

## B.1  Ablation Study on Guidance Sets as Meta-actions

Our framework enables C-MCTD to provide the planner with a guidance set as a meta-action (e.g., $\{0, 0.1, 0.5, 1, 2\}$), rather than a single value. To experimentally isolate the benefit of this design, we compare our default approach against variants that use a single, fixed guidance level.

The results in Table 9 reveal that the optimal guidance level is task-dependent. In PointMaze-Medium, which requires moderately complex planning, higher guidance levels (e.g., 0.5 and 1.0) yield strong performance. Conversely, the more challenging PointMaze-Large environment necessitates a low guidance level (0.1) to achieve a competitive result.

Notably, our default method, which provides the planner with a set of guidance options, achieves robustly high performance across both environments. This demonstrates that the guidance set is a vital mechanism that empowers the planner to adaptively select the most suitable guidance strength for a given task. This adaptability is key to ensuring effective and versatile performance across varying levels of environmental complexity.

## B.2  Ablation Study on Fast Replanning

Fast Replanning is a core component of C-MCTD. To validate its contribution, we conducted an ablation study comparing the performance of the Online Composer with and without this mechanism across various search budgets. The experiments were performed in the maze environments. We omit results from PointMaze Giant as the performance difference was negligible.

As presented in Table 10, the results demonstrate the critical role of Fast Replanning. In all configurations, the version with Fast Replanning consistently outperforms its counterpart. This performance gap becomes more pronounced as the search budget decreases. For instance, in the PointMaze Large environment with a search budget of 50 iterations, our method with Fast Replanning maintains a high success rate of 82%, whereas the variant without it experiences a substantial performance degradation, dropping to 32%.

This analysis confirms that Fast Replanning is not merely an incremental optimization but a crucial element for the robustness and sample efficiency of our tree search algorithm. It enables effective value propagation throughout the search tree, particularly under limited computational budgets.

## B.3  Ablation Study on Stitching Threshold

We investigate the sensitivity of C-MCTD to the stitching threshold hyperparameter, $\epsilon$. In our main experiments, we set $\epsilon = 1.0$, adopting the value from the goal-achievement threshold in the original

Table 11: **The stitching threshold ablation study on PointMaze environments.** Success rate (%) comparison with the variant of Distributed Composer with diverse stitching threshold hyperparameter values, presented as mean $\pm$ standard deviation.

| Environment | Success Rate (%) $\uparrow$ | | | | |
|---|---|---|---|---|---|
| | **5.0** | **2.0** | **1.0** | **0.5** | **0.1** |
| PointMaze Medium | $\mathbf{100}_{\pm 0}$ | $\mathbf{100}_{\pm 0}$ | $\mathbf{100}_{\pm 0}$ | $\mathbf{100}_{\pm 0}$ | $94_{\pm 9}$ |
| PointMaze Large | $65_{\pm 16}$ | $\mathbf{100}_{\pm 0}$ | $\mathbf{100}_{\pm 0}$ | $\mathbf{100}_{\pm 0}$ | $73_{\pm 13}$ |
| PointMaze Giant | $\mathbf{40}_{\pm 24}$ | $26_{\pm 13}$ | $26_{\pm 21}$ | $10_{\pm 10}$ | $0_{\pm 0}$ |

Table 12: **The plan cache ablation study on robot arm cube manipulation tasks.** Run time (sec.) comparison between the versions with and without plan cache, presented as mean $\pm$ standard deviation.

| Method | Run Time (sec.) $\downarrow$ | | | |
|---|---|---|---|---|
| | **single** | **double** | **triple** | **quadruple** |
| OnlineComposer | $\mathbf{19.4}_{\pm 0.3}$ | $50.5_{\pm 1.4}$ | $308.4_{\pm 33.5}$ | $1432.8_{\pm 300.1}$ |
| OnlineComposer with Plan Cache | $\mathbf{19.4}_{\pm 0.2}$ | $\mathbf{45.5}_{\pm 0.7}$ | $\mathbf{166.9}_{\pm 27.1}$ | $\mathbf{242.8}_{\pm 16.1}$ |

MCTD framework [32]. To analyze the robustness of our method to this choice, we conducted an extensive ablation study with our Distributed Composer.

Our analysis, presented in Table 11, reveals that the optimal stitching threshold $\epsilon$ is highly dependent on task complexity. In the PointMaze Medium environment, which represents a simpler planning problem, our method exhibits strong robustness. Optimal performance is maintained across a broad range of $\epsilon$ values (0.5 to 5.0), with only a minor degradation observed at the highly restrictive threshold of $\epsilon = 0.1$. For the more challenging PointMaze Large environment, a clear trade-off emerges. An overly permissive threshold (e.g., $\epsilon = 5.0$) results in spurious stitching, such as incorrectly connecting trajectories across physical barriers, thereby degrading performance. Conversely, an overly restrictive threshold (e.g., $\epsilon = 0.1$) prevents valid connections, impairing the construction of long-horizon solutions. This highlights the existence of an optimal range for $\epsilon$ in moderately complex tasks. Intriguingly, in the most complex PointMaze Giant environment, a more lenient threshold ($\epsilon = 5.0$) yields the best performance. The vast scale of this environment makes connecting distant states inherently difficult. A larger $\epsilon$ increases the likelihood of successful stitching, facilitating the formation of long-range plans. In this scenario, maximizing connectivity, even at the cost of some precision, is more effective than enforcing high-precision stitching that results in sparse or failed connections.

## B.4 Ablation Study on Plan Caching

To quantify the computational efficiency gained by our plan caching method, we conducted an ablation study on robotic manipulation tasks of increasing complexity. The results, summarized in Table 12, measure the wall-clock time required to solve tasks involving one to four cubes.

For tasks with one or two cubes, the runtime difference is minimal, as the planning overhead is low. However, as the task complexity increases, the benefits of plan caching become exponential. This is due to the combinatorial explosion of the planning space in multi-object manipulation. For the four-cube task, our method with plan caching achieves an approximately $6\times$ runtime reduction in runtime compared to the baseline without caching (242.8s vs. 1432.8s). These results confirm that plan caching is an effective strategy for mitigating the computational burden in complex tasks with large, structured planning spaces.

## B.5 Analysis of Preplan Composer's Graph Construction Overhead

The Preplan Composer (PC) variant relies on a one-time, offline graph construction process. In this section, we quantify this computational overhead and contextualize it by comparing the cost to the online planning times of our other methods. We emphasize that this is a pre-inference step performed only once per environment. All timings were measured on the PointMaze environments using 8 NVIDIA GeForce RTX 4090 GPUs.

Table 13: **The comprehensive planning runtime comparison on PointMaze environments.** Planning run time (sec.) and success rate (%) comparison across the methods except CompDiffuser.

| Method | Run Time (sec.) ↓ | | | Success Rate (%) ↑ | | |
|---|---|---|---|---|---|---|
| | **Medium** | **Large** | **Giant** | **Medium** | **Large** | **Giant** |
| Diffuser-Replan | 80.2 | 144.7 | 166.8 | 38 | 40 | 0 |
| Diffuser-DatasetStitch | 8.4 | 8.3 | 9.4 | 32 | 0 | 0 |
| Diffusion Forcing | 18.8 | 26.1 | 30.1 | 53 | 20 | 0 |
| Diffusion Forcing-DatasetStitch | 10.9 | 11.5 | 10.9 | 52 | 26 | 0 |
| SSD | 32.8 | 33.8 | 42.7 | 40 | 40 | 0 |
| MCTD-Replan | 166.5 | 503.7 | 630.1 | 90 | 20 | 0 |
| MCTD-DatasetStitch | 50.2 | 77.2 | 53.8 | 12 | 2 | 0 |
| Online Composer | 83.6 | 91.2 | 269.8 | 93 | 82 | 0 |
| Distributed Composer | 26.7 | 39.1 | 530.7 | 100 | 100 | 26 |
| Preplan Composer | 11.1 | 21.5 | 36.9 | 100 | 100 | 100 |

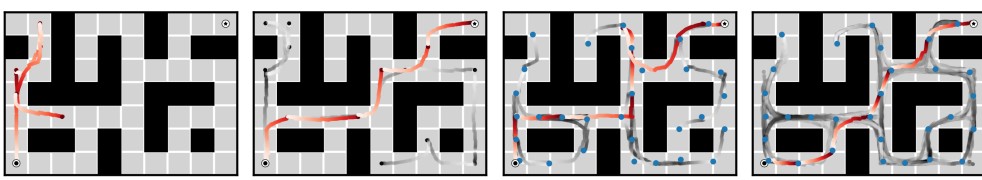

(a) MCTD-Replan     (b) Online Composer     (c) Distributed Composer     (d) Preplan Composer

Figure 4: **Qualitative comparison of stitched planning approaches. (a) MCTD-Replan:** Sequential replanning (trajectories shown from light to dark red) enables plan concatenation but results in myopic, short-sighted paths. **(b) Online Composer:** Plan-level search explores beyond the training data horizon. Gray lines denote discarded candidate plans during the search. **(c) Distributed Composer:** Search efficiency is improved by initiating searches from multiple starting positions (blue dots). Gray lines are unselected plans originating from the starting positions. **(d) Preplan Composer:** A pre-built graph is leveraged to generate higher-quality plans with more optimized efficiency. Gray lines represent pre-computed but unused plans.

We measured the graph construction times for the medium, large, and giant PointMaze maps, which contain 10, 30, and 70 waypoint nodes, respectively. The corresponding construction times were 46.16, 83.18, and 72.69 seconds. Notably, building the graph for the giant environment is faster than for the large one. This counter-intuitive result occurs because the higher density of nodes (70 vs. 30) allows for a shorter maximum search length between any pair of nodes, reducing the per-edge planning complexity.

This one-time cost represents a highly practical initial investment. To put this into perspective, the entire graph construction for the Large environment (83.18s) takes less time than a single planning query with our Online Composer (91.2s, see Table 2). Once this graph is built, it is reused for all subsequent planning tasks, including those with unseen start-goal pairs. Therefore, the initial computational cost is effectively amortized across numerous runs, leading to the significant inference-time efficiency gains of Preplan Composer. This strategic trade-off—a modest offline computation for a substantial and repeated online speedup—is a key advantage of the proposed method.

## B.6 Comprehensive planning runtime comparison

This section provides a comprehensive comparison of the planning runtime for our proposed methods against all baselines, with detailed results presented in Table 13. We note that runtime for CompDiffuser was not reported in the original paper and is therefore excluded from this analysis.

The results underscore that our C-MCTD framework achieves a state-of-the-art trade-off between task success and computational efficiency. In the Medium environment, all C-MCTD variants achieve success rates exceeding 90%. In comparison to the strongest baseline, MCTD-Replan (90% success), our methods are 2-15× faster. This advantage becomes more pronounced in the Large environment,

where our Distributed and Preplan Composers achieve a 100% success rate, a stark contrast to the sub-50% rates of all baselines. Notably, this superior performance is achieved with planning times that are comparable to or faster than most baselines, with the exception of the DatasetStitch variants. In the most challenging Giant environment, the Preplan Composer is the only method to achieve a 100% success rate (versus 0% for all other baselines), while also recording a highly competitive runtime among non-DatasetStitch approaches.

## C  Algorithms

---

**Algorithm 4** Online Composer

---

**Require:** $s_{\text{start}}$: start state, $s_{\text{goal}}$: goal state, $D$: diffusion planner, $H$: plan horizon, $L$: full-plan horizon, $B$: expansion budget, $\mathcal{G}$: a set of guidance sets
**Ensure:** A full trajectory $\tau_{\text{full}}$ from $s_{\text{start}}$ to $s_{\text{goal}}$ or failure
1:  **procedure** ONLINE COMPOSER($s_{\text{start}}, s_{\text{goal}}, D, H, L, B, \mathcal{G}$)
2:      $v_{\text{root}} \leftarrow \text{Node}(s_{\text{start}}, \text{null})$                      ▷ Create root node with start state
3:      $\mathcal{T} \leftarrow (V = \{v_{\text{root}}\}, E = \emptyset)$                         ▷ Initialize tree with root node
4:      expandedNodes $\leftarrow 0$
5:      **while** expandedNodes $< B$ **and** no node close to $s_{\text{goal}}$ **do**
6:          $v \leftarrow \text{SelectNodeToExpand}(\mathcal{T})$                    ▷ Based on UCT value
7:          $G_s \leftarrow \text{SelectGuidanceSet}(v, \mathcal{G})$                  ▷ Select guidance set
8:          $\tau \leftarrow D(v.\text{plan}, s_{\text{goal}}, H, G_s)$                     ▷ Generate guided plan
9:          $\tau_{\text{child}} \leftarrow \text{Concatenate}(v.\text{plan}, \tau)$
10:         reward $\leftarrow \text{FastReplanningSimulation}(\tau_{\text{child}}, s_{\text{goal}}, D, H, L)$
11:         $v_{\text{child}} \leftarrow \text{Node}(\tau_{\text{child}}, \text{reward})$
12:         $\mathcal{T}.V \leftarrow \mathcal{T}.V \cup \{v_{\text{child}}\}$                          ▷ Add child node to tree
13:         $\mathcal{T}.E \leftarrow \mathcal{T}.E \cup \{(v, v_{\text{child}})\}$                      ▷ Add edge to tree
14:         $\text{Backpropagate}(v, \text{reward})$                      ▷ Update ancestors' values
15:         expandedNodes $\leftarrow$ expandedNodes $+ 1$
16:     **end while**
17:     **if** node $v_{\text{goal}}$ exists with $\text{dist}(v_{\text{goal}}.\text{plan}, s_{\text{goal}}) \leq \varepsilon$ **then**
18:         **return** $v_{\text{goal}}.\text{plan}$
19:     **else**
20:         **return** failure
21:     **end if**
22: **end procedure**
23: **procedure** FASTREPLANNINGSIMULATION($\tau_{\text{current}}, s_{\text{goal}}, D, H, L$)
24:     $\tau_{\text{remaining}} \leftarrow \emptyset$                                 ▷ Initialize empty trajectory
25:     $s_{\text{current}} \leftarrow \tau_{\text{current}}[-1]$
26:     **while** length($\tau_{\text{remaining}}$) $< L$ **do**
27:         $\tau_{\text{fast}} \leftarrow D.\text{FastDenoise}(s_{\text{current}}, s_{\text{goal}}, H)$                    ▷ Fast denoising
28:         $\tau_{\text{remaining}} \leftarrow \text{Concatenate}(\tau_{\text{remaining}}, \tau_{\text{fast}})$              ▷ Append trajectory
29:         $s_{\text{current}} \leftarrow \tau_{\text{fast}}[H]$                           ▷ Update current state
30:         **if** no progress or iteration limit reached **then**
31:             **break**
32:         **end if**
33:     **end while**
34:     **return** $\text{EvaluateReward}(\tau_{\text{remaining}})$                    ▷ Return estimated reward
35: **end procedure**

---

**Algorithm 5** Distributed Composer (DC)

---

**Require:** $s_{\text{start}}$: start state, $s_{\text{goal}}$: goal state, $D$: diffusion planner, $H$: plan horizon, $L$: full-plan horizon, $B$: expansion budget, $N$: number of origin positions

**Ensure:** A full trajectory $\tau_{\text{full}}$ from $s_{\text{start}}$ to $s_{\text{goal}}$ or failure

1: **procedure** DISTRIBUTED COMPOSER($s_{\text{start}}, s_{\text{goal}}, D, H, L, B, N$)
2:      $\mathcal{S} \leftarrow \{s_{\text{start}}\} \cup \text{SamplePositions}(N-1)$                 ▷ $N$ origin positions
3:      $G \leftarrow (\mathcal{S}, \emptyset)$                                         ▷ Initialize connectivity graph
4:      expandedTrees $\leftarrow 0$
5:      **while** expandedTrees $< B$ **do**
6:          $s_i \leftarrow \text{SelectOriginPosition}(\mathcal{S}, G)$        ▷ Tree expansion can be implemented in parallel
7:          $\tilde{p}_\theta(\mathbf{x}|s_i) \propto p_\theta(\mathbf{x}|s_i)\exp(\mathcal{J}_\phi(\mathbf{x}))$                 ▷ Define guided distribution
8:          $\mathcal{T}_i \leftarrow \text{GrowTree}(s_i, \tilde{p}_\theta, D, H, L)$             ▷ Grow tree using Online Composer
9:          **for** each $s_j \in \mathcal{S} \setminus \{s_i\}$ **do**
10:              **for** each node $v \in \mathcal{T}_i$ **do**
11:                  **if** $\min_{p \in v.\text{plan}} \text{dist}(p, s_j) < \epsilon$ **then**
12:                      $G.E \leftarrow G.E \cup \{(s_i, s_j, v.\text{plan})\}$          ▷ Add connection to graph
13:                      **break**
14:                  **end if**
15:              **end for**
16:          **end for**
17:          expandedTrees $\leftarrow$ expandedTrees $+ 1$
18:          **if** path exists from $s_{\text{start}}$ to $s_{\text{goal}}$ in $G$ **then**
19:              path $\leftarrow \text{ShortestPath}(G, s_{\text{start}}, s_{\text{goal}})$               ▷ Dijkstra's or A*
20:              $\tau_{\text{full}} \leftarrow \text{StitchPath}(\text{path})$                    ▷ Stitch plans along path
21:              **if** length($\tau_{\text{full}}$) $\leq L$ **then**
22:                  **return** $\tau_{\text{full}}$
23:              **end if**
24:          **end if**
25:      **end while**
26:      **return** failure
27: **end procedure**
28: **procedure** GROWTREE($s_i, \tilde{p}_\theta, D, H, L$)
29:      $v_{\text{root}} \leftarrow \text{Node}(s_i, \emptyset)$                                    ▷ Create root node
30:      $\mathcal{T} \leftarrow (V = \{v_{\text{root}}\}, E = \emptyset)$                           ▷ Initialize tree
31:      **for** $k = 1$ to $K$ **do**                          ▷ K: tree expansion iterations
32:          $v \leftarrow \text{SelectNodeToExpand}(\mathcal{T})$
33:          $\tau \leftarrow D(v.\text{plan}, \tilde{p}_\theta, H)$                ▷ Sample from guided distribution
34:          $\tau_{\text{child}} \leftarrow \text{Concatenate}(v.\text{plan}, \tau)$
35:          reward $\leftarrow \text{FastReplanningSimulation}(\tau_{\text{child}}, s_{\text{goal}}, D, H, L)$
36:          $v_{\text{child}} \leftarrow \text{Node}(\tau_{\text{child}}, \text{reward})$
37:          $\mathcal{T}.V \leftarrow \mathcal{T}.V \cup \{v_{\text{child}}\}$
38:          $\mathcal{T}.E \leftarrow \mathcal{T}.E \cup \{(v, v_{\text{child}})\}$
39:      **end for**
40:      **return** $\mathcal{T}$
41: **end procedure**

---

**Algorithm 6** Preplan Composer (PC) - Building Plan Graph

---

**Require:** $\mathcal{E}$: environment, $D$: diffusion planner, $H$: plan horizon, $N$: number of waypoints
**Ensure:** Plan graph $G$ encoding environment connectivity

1: **procedure** BUILDPLANGRAPH($\mathcal{E}, D, H, N$)       $\triangleright$ Pre-building phase
2:  $S \leftarrow$ SelectWaypoints($\mathcal{E}, N$)     $\triangleright$ Identify $N$ strategic waypoints
3:  $G \leftarrow (S, \emptyset)$           $\triangleright$ Initialize plan graph
4:  **for** each pair $(s_i, s_j) \in S \times S$ where $i \neq j$ **do**
5:    $\tilde{p}_\theta(\mathbf{x}|s_i) \propto p_\theta(\mathbf{x}|s_i) \exp(-\text{dist}(\mathbf{x}, s_j))$  $\triangleright$ Position-based guidance
6:    $\tau_{ij} \leftarrow \text{OC}(s_i, s_j, D, H, \tilde{p}_\theta)$     $\triangleright$ Generate plan
7:    **if** $\tau_{ij} \neq \emptyset$ **then**       $\triangleright$ If connection successful
8:      $G.E \leftarrow G.E \cup \{(s_i, s_j, \tau_{ij})\}$   $\triangleright$ Add to plan graph
9:    **end if**
10:  **end for**
11:  **return** $G$         $\triangleright$ Return completed plan graph
12: **end procedure**

---

---

**Algorithm 7** Preplan Composer - Inference

---

**Require:** $s_{\text{start}}$: start state, $s_{\text{goal}}$: goal state, $G$: prebuilt plan graph, $D$: diffusion planner, $H$: plan horizon, $L'$: local path horizon, $L$: full-plan horizon
**Ensure:** A full trajectory $\tau_{\text{full}}$ from $s_{\text{start}}$ to $s_{\text{goal}}$ or failure

1: **procedure** PC-INFERENCE($s_{\text{start}}, s_{\text{goal}}, G, D, H, L', L$)
2:  **for** each $s_i \in G$ **do**
3:    $\tau_{s \rightarrow i} \leftarrow \text{OC}(s_{\text{start}}, s_i, D, H, L')$
4:    **if** $\tau_{s \rightarrow i} \neq \emptyset$ **then**
5:      $G.E \leftarrow G.E \cup \{(s_{\text{start}}, s_i, \tau_{s \rightarrow i})\}$  $\triangleright$ Add to plan graph
6:    **end if**
7:  **end for**
8:  **for** each $s_i \in G$ **do**
9:    $\tau_{i \rightarrow g} \leftarrow \text{OC}(s_i, s_{\text{goal}}, D, H, L')$
10:    **if** $\tau_{i \rightarrow g} \neq \emptyset$ **then**
11:      $G.E \leftarrow G.E \cup \{(s_i, s_{\text{goal}}, \tau_{i \rightarrow g})\}$  $\triangleright$ Add to plan graph
12:    **end if**
13:  **end for**
14:  path $\leftarrow$ ShortestPath($G, s_{\text{start}}, s_{\text{goal}}$)
15:  **if** path $\neq \emptyset$ **then**
16:    $\tau_{\text{full}} \leftarrow$ StitchPath(path)     $\triangleright$ Compose final solution
17:    **if** length($\tau_{\text{full}}$) $\leq L$ **then**
18:      **return** $\tau_{\text{full}}$
19:    **end if**
20:  **end if**
21:  **return** failure
22: **end procedure**

---

