# OpenReview forum: "Compositional Monte Carlo Tree Diffusion for Extendable Planning"
_NeurIPS.cc/2025/Conference — NeurIPS 2025 spotlight_

### Official Review · Reviewer_y79K · 2025-06-30

**Clarity:** 3
**Significance:** 2
**Originality:** 2
**Rating:** 4
**Confidence:** 2

**Summary:**

This work tackles long-horizon planning tasks, applying methods trained on shorter planning datasets (e.g., upto 100 steps) to test environments that requires longer planning (e.g., 1000 steps). The authors proposed three "stitching" methods to extend planning ability of existing work (Monte Carlo Tree Diffusion). Specifically, the author introduces 1) Online-Composer which performs online MCTS over subplans to extend planning horizon; 2) Distributed Composer to speed up 1); and 3) Preplan Composer which utilizes offline plans to bootstrap online search performance. The authors evaluated their methods against other baselines in various environments in the OGBench, and showed improvements in long-horizon planning against other methods.

**Questions:**

Question:
- Why are baselines different in different experiments? Table 1 included baselines such as CompDiffuser, but they are not included in Table 3 and Table 4?

Typos:
- In L106 the subscripts 1:s and s+1:S are not formatted correctly
- in L205-206 "ParTS-MCTD" is a new method name?

**Ethical Concerns:**

["NO or VERY MINOR ethics concerns only"]

**Final Justification:**

the authors have provided a full runtime table, show clarifying that the proposed methods is both performant and computationally fast. Additionally, the response have clarified a few experimental setting about preplan composer (potentially accessing the test environment in advance). I believe my concerns are mostly addressed, and hence I raise my quality and overall score.

**Limitations:**

yes

**Paper Formatting Concerns:**

I have no formatting concerns.

**Quality:**

3

**Strengths And Weaknesses:**

Strength:
- This paper proposed novel inference-time stitching methods to enables long-horizon planning beyond the length seen during "training".
- The authors experimented with multiple tasks in the OGBench and showed their methods improves generalization in long horizon environments.

Weaknesses:
1. Planning runtime is only compared between the three methods proposed in this work, but not against other prior work. These should be reported in Table 1 along with other methods to provide a more transparent comparison.
2. The best method (Preplan composer) takes significant runtime (mentioned in 228-229) as it involves an additional offline planning phase. This limits generalization ability as 1) the test environment may not be always accessible a priori; and 2) it is unclear how much offline planning is "enough".
3. Preplan composer shows strong performance on giant mazes in Table 1 by utilizing plans composed offline in the same environment. Wouldn't this make comparisons against other planners unfair, since I assume all other planners only had access to small mazes (up to 100steps) during training? Without preplan composer, Table 1 shows that CompDiffuser (a prior work) achieved the best performance compared to the proposed methods.

---

> ### Author Rebuttal · Authors · 2025-07-31
>
> We would like to thank the reviewer for their valuable feedback.
>
> > [W1] Planning runtime is only compared between the three methods proposed in this work, but not against other prior work.
>
> Thank you for highlighting the importance of comprehensive runtime comparisons. We provide planning runtime comparisons with all baseline methods below.
>
> **Comprehensive Planning Runtime Comparison**
>
> We have conducted extensive runtime comparisons across all baseline methods. CompDiffuser's runtime was not reported in their paper and thus could not be included.
>
> **Key Performance-Efficiency Analysis**
>
> Our C-MCTD methods demonstrate superior performance while maintaining competitive planning runtimes:
>
> - **Medium environment**: All C-MCTD variants achieve >90% task success rate. Compared to the strongest baseline MCTD-Replan (90% success rate), our methods **run 2-15× faster.**
> - **Large environment**: Distributed and Preplan Composer achieve 100% task success rate (vs. <50% for baselines) while **maintaining planning times comparable to or better than most baselines**, excluding DatasetStitch variants.
> - **Giant environment**: Preplan Composer achieves 100% task success rate (vs. 0% for all baselines in the table) **with the most competitive runtime among non-DatasetStitch methods.**
>
>
>     | Method | Medium | Large | Giant |
>     | --- | --- | --- | --- |
>     | Diffuser-Replan | 80.2s | 144.7s | 166.8s |
>     | Diffuser-DatasetStitch | 8.4s | 8.3s | 9.4s |
>     | Diffusion Forcing | 18.8s | 26.1s | 30.1s |
>     | Diffusion Forcing-DatasetStitch | 10.9s | 11.5s | 10.9s |
>     | SSD | 32.8s | 33.8s | 42.7s |
>     | MCTD-Replan | 166.5s | 503.7s | 630.1s |
>     | MCTD-DatasetStitch | 50.2s | 77.2s | 53.8s |
>     | Online Composer | 83.6s | 91.2s | 269.8s |
>     | Distributed Composer | 26.7s | 39.1s | 530.7s |
>     | Preplan Composer | 11.1s | 21.5s | 36.9s |
>
> **Significance of Our Contribution**
>
> **C-MCTD introduces efficient inference-time stitching that achieves both effectiveness and efficiency across diverse environments.** The results demonstrate that our tree search-based stitching approach not only outperforms existing methods in task success but does so without prohibitive computational overhead.
>
> We will incorporate this comprehensive runtime comparison into Table 1 of our revised manuscript to ensure transparent comparison across all metrics.
>
> > [W2] The best method (Preplan composer) takes significant runtime (mentioned in 228-229) as it involves an additional offline planning phase. This limits generalization ability as 1) the test environment may not be always accessible a priori; and 2) it is unclear how much offline planning is "enough".
>
> We thank the reviewer for their insightful question regarding the Preplan Composer's offline planning phase and its potential impact on generalization. We would like to clarify two key aspects that address this concern.
>
> First, the offline planning phase does **not** require any prior access to the test environment. Our method, the Preplan Composer, constructs a "plan graph" by generating plans between various representative states. These states are identified by clustering positions solely from the *training dataset*. The diffusion planner then generates these plans between clustered states without any access to the test environment.
>
> Second, we determine the sufficiency of the offline planning using **the same robust reward function employed in the original MCTD framework**. This reward function provides a quantitative measure to evaluate the quality and utility of the pre-generated plans.
>
> This strategic pre-planning is precisely what enables the Preplan Composer to achieve significant efficiency gains at inference time without sacrificing performance or generalizability.
>
> > [W3] Preplan composer shows strong performance on giant mazes in Table 1 by utilizing plans composed offline in the same environment. Wouldn't this make comparisons against other planners unfair, since I assume all other planners only had access to small mazes (up to 100steps) during training? Without preplan composer, Table 1 shows that CompDiffuser (a prior work) achieved the best performance compared to the proposed methods.
>
> We thank the reviewer for raising this important question about the fairness of our experimental setup. We would like to clarify that **all diffusion planners, including ours, were trained under identical conditions** (up to 100 steps on medium/large/giant mazes).
>
> The key distinction of Preplan Composer lies not in its training advantage, but in its **novel inference-time methodology**. Between training and inference, Preplan Composer introduces an additional plan graph building phase that enables more efficient planning during actual deployment. This intermediate step represents a fundamental innovation in our approach—while other methods lack mechanisms to leverage such preprocessing for enhanced planning efficiency, our method demonstrates that inference-time scaling through strategic preprocessing can significantly improve performance.
>
> We acknowledge the reviewer's observation that CompDiffuser achieves the best performance among baselines when excluding Preplan Composer. However, this comparison highlights a crucial distinction: **CompDiffuser requires specifically designed training for compositional planning, whereas C-MCTD achieves superior results through inference-time scaling alone**. This demonstrates that our approach offers a more flexible and generalizable solution that doesn't require retraining the underlying diffusion model.
>
> The strong performance of Preplan Composer on giant mazes validates our core hypothesis that inference-time scaling through tree search-based stitching can overcome the limitations of models trained on shorter trajectories. Rather than being an unfair advantage, this represents the successful realization of our method's design goals.
>
> > [Q1] Why are baselines different in different experiments? Table 1 included baselines such as CompDiffuser, but they are not included in Table 3 and Table 4?
>
> We thank the reviewer for this important question. We would like to clarify our rationale for the baseline selection.
>
> For the benchmarks in **Table 1** (PointMaze, AntMaze), we used the numbers directly reported in the original CompDiffuser paper for a fair comparison.
>
> The benchmarks in **Tables 3 and 4** (Robot Arm Cube Manipulation, Visual PointMaze) were **not evaluated in the CompDiffuser paper**. Implementing and validating their method on our distinct, more complex environments was not feasible during the short rebuttal period due to significant engineering requirements.
>
> We are committed to adding this direct comparison to the revised version to further validate our method's superior performance.
>
> > [Q2] Response to Formatting and Naming Issues
>
> We sincerely thank the reviewer for their meticulous feedback. We have corrected the subscript notation in L106 to ensure mathematical precision. Additionally, we have resolved the naming inconsistency in L205-206; "ParTS-MCTD" was an internal name for our Distributed Composer, and we have updated the manuscript to use this term consistently for clarity.

---

> > ### Comment · Reviewer_y79K · 2025-08-04
> >
> > Thank you for your response. I believe the runtime table and the clarification on preplan composer have addressed my concern. I have raised the quality score and overall score.

---

> > > ### Author Response · Authors · 2025-08-05
> > >
> > > Thank you for the opportunity to address your concerns. The revised manuscript will be updated to reflect these points. We sincerely appreciate your insightful comments, which have significantly strengthened our paper.

---

### Official Review · Reviewer_rpAX · 2025-07-01

**Clarity:** 2
**Significance:** 3
**Originality:** 3
**Rating:** 5
**Confidence:** 4

**Summary:**

This paper try to address the limited planning horizon issue of MCTD framework by adding hierarchical planning strategy that stitch the plans in various different ways. It achieved substantially better performance or efficiency in out-of-domain long-horizon maze navigation tasks compared with Diffuser and MCTD.

**Questions:**

* One of the most important question is what is the formulation of the problem that C-MCTD works the best among the Diffuser, MCTD, traditional A* or MCTS algorithms such as UCT, PUCT, GumbelMuzero, etc. Where could we apply this algorithm in practice? What is the supporting evidence?
* If applicable, I am curious about the performance of A* search with manhattan distance heuristics in pointmaze problem, including the runtime and successrate.
* Looks like the MCTD and C-MCTD assume there is no need to use transition function to simulate the actual state changes for action execution? Are the environments in the experiment deterministic in transition? Will there be any impact if the environment has a highly random transition function?
* In POMDP tasks, are you conducting the search offline as well and replan online? Is the low-performance is caused by no transition function's simulation, so that the gap between actual execution and plan is big?

**Ethical Concerns:**

["NO or VERY MINOR ethics concerns only"]

**Final Justification:**

I personally believe it is a good paper if the author could address the issues mentioned in the rebuttal during the revisions:

1. The inherent limitations of traditional algorithms (PRM, MCTS, etc), coupled with a clear problem formulation in the preliminaries section, to highlight the distinct advantage of C-MCTD. Also, reflect the formulation on the experimental setup.
2. Online planning or planning in a stochastic environment.
3. Assumptions made in different variants of C-MCTD.

My main reason for acceptance is that
1. It indeed has good reported performance in long-horizon planning tasks, and also
2. provides reasonable solutions to the extensive run-time issues;
3. The assumption is also reasonable, as we generally expect the training dataset to be fairly similar to the testing cases.

Thus, with clear assumptions and a discussion of the limitations, I believe this research is a meaningful attempt to address long-horizon planning problems.

**Limitations:**

I think in Discussion section, the author compare the limitations of different variants of C-MCTD, but did not actually discuss the limitation of C-MCTD compared with MCTD, Diffuser, and traditonal search algorithms. This should be included with supporting evidence to better define the scope of the application of C-MCTD.

**Paper Formatting Concerns:**

No formatting issues.

**Quality:**

3

**Strengths And Weaknesses:**

**Strength**

1. The overall performance is substantially strong in long-horizon maze navigation tasks compared with MCTD and Diffuser.
2. The overall framework seems to be general. The paper applied the idea in both the maze navigation and robot arm cube manipulation.

**Weaknesses**

1. The algorithm could become quite complex due to the inherent computational cost of MCTS. In the worst case, UCT has an exponential complexity of the form $\Omega(\exp(\exp(\exp(...\exp(1)))))$ (depth -1 times of $\exp$) [1], and the actual worst-case performance could be even more prohibitive in practice when composing multiple UCT instances and diffusion process.
2. If understand correctly, the algorithm seems to tackle the offline planning problems. However, MCTS is a online planning algorithm because of its anytime property and effectivenness in online MDP/POMDP (POMCP or DESPOT for POMDP). Given the essential slow runtime of diffusion, the proposed method could be very limited in real-time or low-latency planning scenarios.
3. The paper might need a more comprehensive discussion of practical application scenarios where the proposed C-MCTD method offers a distinct advantage. For example, determinisitic pointmaze navigation could be tackled effectively by A* search. As for POMDP, there are also mature algorithms such as POMCP and DESPOT. It would strengthen the paper to include a discussion on when and why C-MCTD should be preferred over these traditional approaches, and under what conditions it significantly outperforms them.

[1] Qoquelin & Munos, Bandit Algorithms for Tree Search, UAI 2007

---

> ### Author Rebuttal · Authors · 2025-07-31
>
> We thank the reviewer for their constructive comments.
>
> > [W1] On Computational Complexity and Our Proposed Solutions
>
> We thank the reviewer for their insightful comment regarding the computational complexity of MCTS. The reviewer is correct that the worst-case exponential complexity of MCTS is a critical consideration and it has been figured out in the previous work, MCTD. Indeed, addressing this inherent challenge was one of primary motivations for our work.
>
> Our proposed method, C-MCTD, is designed specifically to mitigate this complexity through two key innovations: the **Distributed Composer** and the **Preplan Composer**.
>
> 1. **Distributed Composer:** This approach prevents the search space from exploding in deep trees by parallelizing the search from multiple origin points, effectively managing the search depth.
> 2. **Preplan Composer:** This method further enhances efficiency by pre-generating and caching the plans between the tree origin positions into a graph, significantly reducing the online search burden at inference time.
>
> The effectiveness of these solutions is empirically validated in our experiments. As shown in **Table 2**, compared to the Online Composer, our Distributed Composer achieves substantial speedups of ~3× and ~2× on Medium and Large maps, respectively. Furthermore, the Preplan Composer demonstrates even more significant gains, with speedups of ~7× and ~4×.
>
> > [W2] On the Practicality for Online Planning and the Role of the Preplan Composer
>
> We thank the reviewer for their insightful comment. We agree that the inherent computational cost of diffusion models presents a significant challenge for their application in online planning. Addressing this very challenge is one of central motivations for our work.
>
> The Preplan Composer variant is designed precisely to overcome the runtime limitations mentioned by the reviewer. By pre-generating a graph of potential plans in an offline phase, it shifts the computational burden away from the time-critical online execution. During the online phase, the agent primarily performs efficient lookups and minimal stitching, rather than expensive generative planning.
>
> As empirically demonstrated in Table 2, this approach is highly effective. **The Preplan Composer achieves a 4-7x speedup in planning time compared to Online Composer and 2x speedup compared to Distributed Composer that generate plans entirely online**, showcasing its viability for scenarios where low latency is crucial. Therefore, our work provides a practical pathway to harness the power of diffusion models for complex online planning tasks.
>
> > [W3 & Q2] Clarification on the Task Complexity and the Advantage of C-MCTD
>
> We thank the reviewer for their insightful suggestion to elaborate on the practical application scenarios for our proposed C-MCTD. We respectfully clarify that the environments evaluated in our work present significant challenges that render traditional search algorithms, such as A* or POMDP solvers, impractical.
>
> For instance, the PointMaze navigation task features continuous state and action spaces. An optimal agent navigating the giant map requires a trajectory of at least 500 steps. Applying conventional search algorithms to such a long-horizon task in a continuous space would involve exploring a prohibitively large, if not intractable, search space. Our method, C-MCTD, is precisely designed to tackle these complexities by leveraging diffusion models for tree search-based stitching of short trajectories, a task for which methods like A* are not suited.
>
> We acknowledge that a more explicit discussion of these task characteristics will strengthen our paper. In the revised version, we will dedicate a section to discuss the limitations of traditional approaches in our problem settings.
>
> > [Q1] Clarification on the Optimal Use Cases and Practical Applications of C-MCTD
>
> Thank you for the insightful question, which clarifies where C-MCTD is most effective.
>
> C-MCTD is specifically designed to excel in scenarios where diffusion models are constrained by limited training data length and thus cannot generate sufficiently long, high-quality plans. Our method's core strength lies in its ability to effectively stitch together short trajectories through a tree-search mechanism at inference time, enabling long-horizon planning that surpasses the limitations of the training data.
>
> The supporting evidence for this is presented in our experimental results.
>
> - **As shown in Table 1,** when the required planning horizon is longer than the training trajectories, C-MCTD outperforms all baseline methods—including Diffuser, Diffusion Forcing, and their variants using replanning or stitched datasets—**by a substantial margin of 10-100%.**
> - **Furthermore, Table 3 highlights** that C-MCTD's advantage is even more pronounced (achieving 10-100% better performance) in tasks that necessitate compositional planning, **directly validating the efficacy of our tree-search-based stitching.**
>
> Regarding traditional search algorithms, they are often not well-suited for the complex tasks we address, which involve continuous state-action spaces and demand very long-horizon planning.
>
> > [Q3] On the Role of the Transition Function and Environmental Stochasticity
>
> We thank the reviewer for their insightful question regarding our model's assumptions and its applicability to stochastic environments.
>
> **1. Clarification on the Transition Function:** The reviewer is correct in observing that our methods, MCTD and C-MCTD, do not require an explicit transition function $T(s,a,s′)$. Our framework leverages a diffusion model trained on an offline dataset of short trajectories. By doing so, our model directly learns the underlying distribution of valid state sequences, implicitly capturing the dynamics demonstrated in the dataset. **This sequence-level modeling is a core design choice that allows our method to generate coherent long-horizon plans without the need for step-wise state transition modeling.**
>
> **2. On Environmental Stochasticity:** We confirm that the environments used in our current experiments are deterministic. This was a deliberate choice to first establish a strong foundation for diffusion-based planning and demonstrate the effectiveness of our core contribution: inference-time tree search and stitching.
>
> The reviewer raises an excellent point about performance in highly stochastic environments. A direct application of our current model would likely result in learning an "averaged" plan from the varied outcomes in the dataset, which may degrade planning quality.
>
> Extending our framework to such settings is a promising direction for future work. We will explicitly clarify the deterministic scope of our experiments and **add this important discussion on stochasticity as a future work direction in the revised manuscript.**
>
> > [Q4] Clarification on the Primary Cause of Performance Degradation in POMDPs
>
> We appreciate the reviewer's insightful question regarding performance in POMDP settings. While the gap between actual execution and planning due to the absence of transition function simulation could indeed contribute to performance degradation, our analysis reveals a more fundamental issue.
>
> As documented in the MCTD paper (Appendix A7.7) and confirmed through our extensive experiments, the primary challenge stems from **trajectory collapse** during long-horizon generation in POMDP environments—specifically, later trajectories becoming stuck in suboptimal states. The MCTD paper attempted to address this through frequent online replanning at short intervals; however, this approach remains inherently **myopic** and constrains performance improvements.
>
> Our C-MCTD framework directly tackles this limitation through **inference-time stitching methods** that maintain global coherence while avoiding trajectory collapse. This approach demonstrates substantial performance gains, particularly in complex large-scale environments where traditional replanning strategies struggle.
>
> > [L] On the Limitations and Application Scope of C-MCTD
>
> We thank the reviewer for the constructive suggestion to further elaborate on the limitations of our method. We will clarify this in the final version.
>
> Our discussion did not originally center on a direct comparison with standard MCTD or Diffuser because **C-MCTD is architected to address a fundamentally different challenge: composing long-horizon plans that exceed the trajectory lengths of the training data.** In contrast, methods like MCTD and Diffuser focus on optimizing planning *within* the horizon of the training data. C-MCTD is a solution for inference-time generalization to out-of-distribution plan lengths, a scope where direct application of standard diffusion planners is not feasible.
>
> When compared to traditional search algorithms, a key prerequisite for C-MCTD—and diffusion planners in general—is the requirement for an offline trajectory dataset and a corresponding offline model training phase. For our Preplan Composer variant, an additional preliminary step involves the construction of a plan graph.
>
> Therefore, the application scope of C-MCTD is specifically for domains where: 1) an offline dataset is available, and 2) the primary goal is to solve long-horizon tasks that surpass the lengths of the available demonstration trajectories.
>
> Within this defined scope, C-MCTD demonstrates significant value.
>
> - It substantially outperforms relevant baselines, as evidenced in **Table 1**.
> - It proves effective in complex compositional tasks that inherently demand long-term planning, such as the multi-cube robotic manipulation task (**Table 3**) and the Visual PointMaze task, where generating long plans is notoriously difficult (**Table 4**).
>
> We will add this detailed discussion to the paper to clearly define the application scope and limitations of C-MCTD.

---

> ### Comment · Reviewer_rpAX · 2025-08-01
>
> I sincerely appreciate the author's detailed feedback. However, I am still confused about the following points.
>
> **1. Compare with traditional search algorithms**
>
> The author claimed that C-MCTD is effective for large continuous spaces and large action spaces, and uses the point-maze as a testbed for experimental demonstration. However, given that the problem is fully observable, a more straightforward approach is to use the probabilistic roadmap algorithm (PRM) [1] or its variants like [2] with A*, which is the foundation of robot planning in very high-dimensional, continuous configuration spaces and can be very efficient. There are numerous methods, e.g., RRT [3], that are well-established in robot motion planning for fully observable, continuous configuration spaces.
>
> Thus, the assumption that the A* is not suitable for the large, continuous spaces maze problem is not valid. The author should think more carefully about when to use traditional algorithms, such as PRM + A*, and when to use C-MCTD.
>
> [1] Probabilistic roadmaps for path planning in high-dimensional configuration spaces
>
> [2] The Bridge Test for Sampling Narrow Passages with Probabilistic Roadmap Planners
>
> [3] Sampling-based Algorithms for Optimal Motion Planning
>
> **2. Online planning**
>
> The proposed variants, such as the preplan composer, indeed have a substantial improvement in runtime efficiency compared with the naive C-MCTD. However, the latency is still very high for many online planning tasks, such as robot online motion planning. The need for online planning arises because the environment is constantly changing, and planners must adjust their plans in real-time to keep up with these changes. It is equivalent to a stochastic transition function. The current latency still cannot meet the requirement due to the use of the diffuser, which is inherently slow. Additionally, the underlying assumption of preplan is that the testing problems are similar to those in the training set. If the new environment is substantially different, most pre-computed plans are invalid, and it will not work. This leads to the inherent limitation.
>
> I understand that the paper cannot solve everything and must make a few assumptions. Thus, I suggest the author primarily focus on addressing the 1st concern, and can have more discussion on the second concern in the paper. The 1st concern is more severe, as after the rebuttal, it has become further unclear about the distinct advantage of C-MCTD.

---

> > ### Author Response · Authors · 2025-08-03
> > **Reply to the comment**
> >
> > We sincerely thank the reviewer for the thoughtful feedback and for engaging deeply with our work. The comments have been invaluable in helping us clarify the contributions and positioning of our paper.
> >
> > ### **On the Comparison with Traditional Search Algorithms**
> >
> > We thank the reviewer for raising this crucial point. We acknowledge that our initial discussion of traditional search algorithms was not sufficiently nuanced. The reviewer is correct that planners like PRM and RRT, when paired with search algorithms such as A*, are highly effective and represent the state-of-the-art for motion planning in fully observable, continuous spaces.
> >
> > When comparing with these traditional search algorithm (e.g., PRM+A*), the primary distinction of our approach lies in the **problem setting**: our work is framed within the **offline goal-conditioned reinforcement learning** context. In this paradigm, the agent learns from a fixed dataset of trajectories and **does not have access to the environment for active interaction** during the planning phase.
> >
> > This constraint fundamentally differentiates our method from traditional planners:
> >
> > - **Traditional Planners (e.g., PRM, RRT):** These methods iteratively construct a graph (or roadmap) by actively sampling the configuration space and querying the environment to validate connections (i.e., performing collision checks). This requires continuous, online access to the environment simulator or the physical world.
> >
> > - **Our Preplan Composer:** In contrast, our method constructs a planning graph **without any environment interaction during the graph-building process**. It leverages a diffusion model trained solely on the offline trajectory data. The model learns the underlying patterns of valid movements from the data and generates a graph of high-quality paths.
> >
> > Therefore, a direct comparison shifts from search algorithms alone to the **methods of graph construction under different environmental access assumptions**. While PRM/RRT build graphs from active exploration, our Preplan Composer builds a graph from passive, offline data.
> >
> > From this perspective, our Preplan Composer can be seen as a new approach to adapt classical planning principles to the offline goal-conditioned RL setting. We are grateful for the opportunity to clarify this, and we will add a detailed discussion comparing these approaches and their underlying assumptions to the revised paper.
> >
> > ### **On Online Planning and Latency**
> >
> > We concur with the reviewer regarding the current latency of C-MCTD and its implications for real-time applications. Although the Preplan Composer significantly improves efficiency over the online variants, we acknowledge that the inference time does not yet meet the stringent requirements of many online planning tasks.
> >
> > We also agree with the reviewer’s assessment of the generalization limits. This is a shared limitation of diffusion planners. Indeed, performance is contingent on the similarity between the training and testing environments; if a new environment differs substantially, the pre-trained planner may generate invalid plans.
> >
> > In the revised manuscript, we will explicitly incorporate a more thorough discussion of these current limitations. Furthermore, as a path toward addressing the latency issue, we will investigate and cite recent work on accelerating inference-time scaling diffusion-based planners, such as the parallel sparse planning approach presented in [1]. We thank the reviewer for pushing us to address these practical and important aspects of our work.
> >
> > [1] Fast Monte Carlo Tree Diffusion: 100x Speedup via Parallel Sparse Planning, Yoon et al., arXiv:2406.09498

---

> > > ### Comment · Reviewer_rpAX · 2025-08-03
> > >
> > > Thank you for your elaboration and clarification. All my concerns are addressed.
> > >
> > > I suggest the author modify the paper and add an in-depth discussion on:
> > >
> > > 1. The inherent limitations of traditional algorithms (PRM, MCTS, etc), coupled with a clear problem formulation in the preliminaries section, to highlight the distinct advantage of C-MCTD. Also, reflect the formulation on the experimental setup.
> > > 2. Online planning or planning in a stochastic environment.
> > > 3. Assumptions made in different variants of C-MCTD
> > >
> > > I lean towards acceptance. Good work and all the best.

---

> > > > ### Author Response · Authors · 2025-08-03
> > > >
> > > > We sincerely thank the reviewer for their valuable feedback and support for our work. We are happy to have addressed all your concerns.
> > > >
> > > > As suggested, our final revision will feature a more in-depth discussion on the limitations of traditional algorithms, supported by a clearer problem formulation to highlight C-MCTD's advantages. We will also expand on the topic of online planning in stochastic environments and explicitly state the assumptions for each variant of our method.
> > > >
> > > > We are grateful for your guidance in strengthening our paper.

---

### Official Review · Reviewer_Q2ax · 2025-07-03

**Clarity:** 3
**Significance:** 3
**Originality:** 2
**Rating:** 4
**Confidence:** 3

**Summary:**

This work presents an extension of the Monte Carlo Tree Diffusion (MCTD) approach for long-horizon planning, termed C-MCTD. The proposed method enables the construction of long-horizon plans by effectively "stitching" together relatively short plans generated by MCTD. The authors introduce three strategies to achieve this—online, distributed, and preplan—and apply them to long-horizon maze navigation and multi-cube robot arm manipulation tasks.

**Questions:**

- How is the guidance set defined and implemented in the proposed method? Despite frequent references to "guidance" in the main text and appendix, its precise formulation and implementation remain unclear.
- What is the size of the subgoal set, and how are subgoals sampled during planning? Is randomly scattering subgoals a valid strategy for complex long-horizon tasks, or could this limit planning efficiency and interpretability?
- Could the authors provide further details on the hyperparameters used in the experiments? For instance, how was the epsilon threshold for stitching trajectories determined, and was the method robust to smaller epsilon values during execution?
- (minor) spacing in Table 3 : Online Composer

**Ethical Concerns:**

["NO or VERY MINOR ethics concerns only"]

**Final Justification:**

Having considered both the initial submission and the authors’ rebuttal, I believe that the work has reached the standard expected for a NeurIPS conference paper. Accordingly, I have adjusted my evaluation from a borderline reject to a borderline accept.

My primary concerns were related to the lack of detailed explanations of the experiments and the justification of the reported results. The authors have addressed many of these concerns in their response. However, I expect the final version of the manuscript to include the detailed clarifications and result visualizations as outlined in the rebuttal.

**Limitations:**

yes

**Quality:**

3

**Strengths And Weaknesses:**

### Strengths
- Divide-and-conquer approach for long-horizon tasks: This work builds upon the idea of MCTD by proposing a method that enables planning over longer horizons than the trajectory lengths supported by MCTD. By introducing and comparing three different composition strategies in C-MCTD, the authors allow for flexible selection of an appropriate strategy depending on the task requirements.
- Clear explanation of some key algorithmic components: The differences among the proposed composition strategies are well explained, particularly in terms of the components that distinguish each approach. The inclusion of algorithmic details in the appendix and the release of open-source code contribute to the reproducibility of the proposed method.

### Weaknesses
- Shallow analysis of candidate tree origin position proposal : While the proposed method extends MCTD by composing multiple sub-trajectories via tree search—similar to hierarchical RL—it lacks in-depth discussion on the generation and maintenance of subgoals, which are crucial in hierarchical planning. The method treats tree origin positions as subgoals but does not provide substantial analysis on how these subgoals should be selected or maintained. In particular, for the distributed and preplan composers, subgoals are either randomly sampled or derived from previous trajectories, without further discussion on their semantic significance or strategic role.
- Lack of clarity regarding the use and definition of the guidance set: Despite the frequent use of the term "guidance" throughout the manuscript and appendix, there is a lack of concrete mathematical formulation or detailed explanation regarding how the guidance set is actually utilized. A more explicit clarification—either in the main text or at least in the appendix—is necessary.
- Insufficient discussion of the preplan composer: The authors claim that the preplan composer achieves 100% accuracy, which is not necessarily an incorrect observation. However, the emphasis on this result raises questions about the true contribution of the preplan composer. Without deeper analysis on what kinds of preplans are effective for long-horizon or multi-stage tasks—and how these preplans contribute to improved efficiency or accuracy—the method risks being perceived as a simple concatenation of ground-truth trajectory fragments.
- Limited empirical validation of the proposed method: The model is only evaluated on two tasks (maze navigation and cube arrangement). While the results on these tasks demonstrate some level of robustness, the general applicability of the method remains uncertain. Additional experiments across diverse environments would be necessary to demonstrate that the proposed method is not overly specialized to these specific scenarios.
- Lack of visualization for experimental results: The only visual comparison presented is the conceptual illustration in Figure 1. To better understand the behavior and differences between the proposed composition strategies, it would be helpful to include visualizations of the actual trajectories generated in the experiments.

---

> ### Author Rebuttal · Authors · 2025-07-31
>
> We thank the reviewer for their constructive comments.
>
> > [W1] Shallow analysis of candidate tree origin position proposal
>
> We thank the reviewer for their insightful feedback and for highlighting the need for a clearer exposition of our subgoal selection strategy. We respectfully wish to clarify that the selection of tree origin positions (subgoals) for our Distributed and Preplan Composers is a principled, data-driven process, not a random one.
>
> **As detailed in the Appendix (Lines 885-890)**, we identify these subgoals by performing k-means clustering on the trajectories from the training dataset. The centroids of these clusters represent strategically significant and diverse states—regions that are frequently traversed or serve as critical waypoints in the environment's state space. This data-driven approach ensures that our subgoals possess **semantic significance**, directly reflecting the structure of the underlying environment. **They are not arbitrarily chosen but are representative of key decision points.**
>
> We acknowledge that this crucial detail was not sufficiently emphasized in the main body of the paper. To address this, we will revise the manuscript.
>
> > [W2] Lack of clarity regarding the use and definition of the guidance set
>
> We thank the reviewer for their insightful comment and for highlighting the need for a more explicit definition of the "guidance set." We acknowledge that the role of this component could be further clarified to better underscore its importance within our framework.
>
> The term "guidance" in our work refers to the steering mechanism inherent in our classifier-guided approach, which is directed by the value function $\mathcal{J}_\phi(\mathbf{x})$ as formulated in Equation (2).
>
> The **guidance set** is a collection of hyperparameter weights that modulate the strength and nature of this guidance. Crucially, our inference-time scaling planner takes this set as an argument during the node expansion phase of C-MCTD. **This mechanism empowers the planner to generate effective and diverse local plans by dynamically adjusting the influence of the value function at each step of the tree search.** We studied the ablations for the guidance set in the rebuttal for the reviewer ECBR's review, `Ablation Study on the Efficacy of the Guidance Sets`.
>
> To address the reviewer's valid point, we will incorporate these clarifications into the revised manuscript.
>
> > [W3] Insufficient discussion of the preplan composer
>
> We thank the reviewer for their insightful comment, which highlights the need for a more detailed discussion of the Preplan Composer. We would like to clarify its mechanism and contribution, addressing the concern that it might be a "simple concatenation of ground-truth trajectory fragments."
>
> The Preplan Composer is a method designed to amortize the computational cost of planning, enabling highly efficient inference for long-horizon tasks. Its contribution is not merely achieving high accuracy but in structuring the planning problem to drastically reduce online computational overhead. The process unfolds in three distinct stages:
>
> 1. **Offline Plan Graph Construction:** First, we construct a "plan graph" entirely offline. This involves generating plans between a set of diverse "tree origin positions" sampled from the training data distribution. This computationally intensive step is performed only once and the resulting graph can be reused across various tasks within the same environment.
> 2. **Inference-Time Graph Connection:** At inference time, given a novel start-goal pair, our system generates only short "bridging" plans to connect the start and goal to their nearest nodes in the pre-computed plan graph. Since the graph is dense, these bridging plans are typically very short, requiring minimal online search.
> 3. **Final Path Assembly:** Finally, we perform a rapid search (e.g., Dijkstra's algorithm) on the graph to find the shortest path of nodes connecting the start and goal. The final plan is an assembly of the pre-generated plans corresponding to the edges in this path.
>
> Crucially, each segment in this final path is a plan **generated by our learned diffusion planner**, not a ground-truth trajectory fragment. The Preplan Composer's novelty lies in its strategic pre-computation, which amortizes the planning effort. This allows our method to efficiently solve complex, long-horizon problems at inference time, a significant step beyond methods that perform the entire search online.
>
> > [W4] Limited empirical validation of the proposed method
>
> **Our primary objective was to directly address a critical and known limitation of prior work (MCTD)**: its performance degradation in settings constrained by short training trajectories. To create the most rigorous and fair comparison, **we intentionally selected the very environments where MCTD had previously demonstrated its effectiveness.**
>
> While we concur that evaluation on a broader set of tasks is a valuable direction for future work, our focused experiments provide the necessary and compelling evidence that our proposed method resolves a significant bottleneck in state-of-the-art inference-time scaling planning algorithms.
>
> > [W5] Lack of visualization for experimental results
>
> We thank the reviewer for this valuable feedback. We concur that visualizing the experimental results will significantly improve the paper. We will update them in our revised version.
>
> > [Q1] How is the guidance set defined and implemented in the proposed method?
>
> We thank the reviewer for the opportunity to clarify our method.
> The term "guidance" denotes the steering of the generation via a value function (Eq. 2). The "guidance set" is the concrete implementation of this principle: a collection of predefined weights that control the strength of this steering.
> We will ensure this definition is stated more explicitly in the final version of the paper.
>
> > [Q2] Clarification of Subgoal Set
>
> We thank the reviewer for this insightful question, which allows us to clarify the role and selection of our tree origin positions. We infer that the "subgoals" mentioned in the comment refer to what our paper calls "tree origin positions," which serve as starting points for Distributed and Preplan Composers in our C-MCTD framework.
>
> The size of the origin position set is tailored to the complexity of the environment. For example, we use 10, 30, and 70 positions for the `medium`, `large`, and `giant` maps, respectively. Crucially, these positions are not randomly scattered. **We employ a principled selection strategy to ensure broad and meaningful coverage of the state space.** Specifically, we use k-means clustering on a large set of collected states to identify diverse and representative locations. This deliberate placement is vital for the agent's ability to solve a wide variety of unseen start-goal tasks. **They are not randomly sampled, therefore this does not limit the planning efficiency and interpretability.**
>
> > [Q3] how was the epsilon threshold for stitching trajectories determined, and was the method robust to smaller epsilon values during execution?
>
> We sincerely thank the reviewer for their insightful comment regarding the sensitivity of our hyperparameters. In our experiments, the epsilon threshold for trajectory stitching in the PointMaze environment was initially set to $\epsilon=1.0$. We adopted it directly from the goal-achievement threshold used in the reward function of the original MCTD framework, upon which our work builds.
>
> To explicitly address the reviewer's question about robustness, especially concerning smaller epsilon values, we performed a thorough ablation study using our Distributed Composer. The results, showing the success rate (mean ± std) across different epsilon values, are summarized below:
>
> | Environment | 5.0 | 2.0 | 1.0 | 0.5 | 0.1 |
> | --- | --- | --- | --- | --- | --- |
> | PointMaze Medium | 100±0% | 100±0% | 100±0% | 100±0% | 94±9% |
> | PointMaze Large | 65±16% | 100±0% | 100±0% | 100±0% | 73±13% |
> | PointMaze Giant | 40±24% | 26±13% | 26±21% | 10±10% | 0±0% |
>
> Our analysis of this data reveals a clear relationship between task complexity and the optimal epsilon value:
>
> - **Robustness in Simpler Environments:** For the `PointMaze Medium` environment, which requires relatively straightforward planning, our method demonstrates high robustness. Performance remains optimal across a wide range of epsilon values (from 0.5 to 5.0), only degrading slightly at a very restrictive ϵ=0.1.
> - **A "Sweet Spot" in Complex Environments:** In the more challenging `PointMaze Large` environment, the results highlight a clear trade-off. An excessively large epsilon (e.g., 5.0) degrades performance by causing spurious stitching—for instance, incorrectly connecting trajectories across physical barriers. Conversely, an overly restrictive epsilon (e.g., 0.1) prevents valid stitching opportunities, thus hampering the agent's ability to form long, successful plans. This demonstrates that an appropriately tuned epsilon is beneficial for complex tasks.
> - **The Benefit of Lenient Stitching in Gaint Environments:** Intriguingly, for the `PointMaze Giant` environment, a larger epsilon threshold ($\epsilon=5.0$) yields a noticeable performance improvement. The immense scale of this environment makes it difficult for the planner to connect distant states. A more lenient stitching condition (larger $\epsilon$) facilitates the creation of long-range plans by increasing the frequency of stitching, even if some connections are imprecise. This proves more effective than sparse, high-precision stitching in such scenarios.
>
> We are grateful to the reviewer for raising this important point. This detailed hyperparameter analysis and the accompanying discussion will be incorporated into the revised manuscript.
>
> > [Q4] (minor) spacing in Table 3
>
> We sincerely thank the reviewer for catching the spacing issue in Table 3. We will fix it in our revision.

---

> > ### Comment · Reviewer_Q2ax · 2025-08-04
> > **Official Comment by Reviewer Q2ax**
> >
> > I thank the authors for their response. Many of the concerns I previously raised have been resolved. While some issues remain—particularly regarding detailed explanations of the experimental methods and the visualization of the results—I have decided to revise my score after considering the other reviews in a comprehensive manner.

---

> > > ### Author Response · Authors · 2025-08-05
> > >
> > > We are deeply grateful for the reviewer's insightful comments and for their revised assessment. In the revised version, we will be sure to expand the explanation of our experimental methodology and improve the visualization of our results, as recommended. Your feedback has been instrumental in enhancing the clarity and quality of our work.

---

### Official Review · Reviewer_NuJi · 2025-07-03

**Clarity:** 2
**Significance:** 2
**Originality:** 2
**Rating:** 4
**Confidence:** 2

**Summary:**

This paper proposes Compositional Monte Carlo Tree Diffusion (C-MCTD), a  framework that extend MCTD's planning capabilities beyond training trajectory lengths through three variants, online Composer, Distributed Composer, and Preplan Composer.

The key idea is to stitch multiple short-horizon plans together using tree search at the plan level rather than the subplan level. Online Composer performs sequential plan stitching with hierarchical tree search, Distributed Composer uses parallel trees with strategic connections, and Preplan Composer leverages prebuilt plan graphs. The authors evaluate on maze navigation, robot manipulation, and visual planning tasks, showing improvements over replanning baselines.

**Questions:**

Why not compare against recent methods that specifically address training length limitations, such as hierarchical planning approaches or other compositional methods? The current baselines seem weak.

**Ethical Concerns:**

["NO or VERY MINOR ethics concerns only"]

**Final Justification:**

Reviewer has addressed some of my concerns

**Quality:**

2

**Strengths And Weaknesses:**

1. The paper addresses a real limitation of MCTD - its inability to generate plans longer than training trajectories. The hierarchical approach of searching over plans rather than subplans is reasonable.

2.The core contribution lacks novelty. Plan stitching is well-established, but,,,  the paper essentially applies existing tree search techniques at a higher level.
3.The experimental evaluation has several issues. The baselines are mostly weak replanning methods rather than SoTA long-horizon planners. The comparison lacks recent methods that directly address training length limitations. Results are inconsistent across environments - Distributed Composer sometimes outperforms Online Composer, sometimes doesn't, with no clear explanation.

---

> ### Author Rebuttal · Authors · 2025-07-31
>
> We would like to thank the reviewer for their valuable feedback.
>
> > [W1] The core contribution lacks novelty. Plan stitching is well-established, but,,, the paper essentially applies existing tree search techniques at a higher level.
>
> We sincerely thank the reviewer for their time and valuable feedback. We appreciate the opportunity to clarify the core contributions of our work, C-MCTD. We respectfully posit that our framework offers significant novelties beyond the high-level application of existing tree search techniques. Below, we detail the distinct innovations within each of our proposed composers.
>
> **1. The Online Composer: A Generalization of MCTD**
>
> While building on tree search principles, the Online Composer introduces a novel formulation. It employs a guidance set to define **meta-actions**, which allows each node within the search tree to function as an independent, recursively-scaled planner.
>
> This design makes our Online Composer a **generalization of MCTD**. Specifically, MCTD can be viewed as a special case of our method where planning is constrained only to the root node. By enabling every node to be a planner, C-MCTD can tackle more complex, hierarchical problems.
>
> Furthermore, we introduced a practical innovation, the **Plan Cache**, for the Robot Arm Cube Manipulation task. As shown in the Appendix (Table 5), this mechanism enables highly efficient combinatorial search by reusing successful local-plans. This is a capability that standard high-level plan stitching does not support.
>
> **2. The Distributed Composer: Efficient Parallel Search**
>
> Distinct from conventional tree search, the Distributed Composer initiates **parallel searches from multiple, diverse root nodes** simultaneously by merging these parallel search trees, which is guided by a latent distance heuristic.
>
> This approach effectively partitions the overall search space. By preventing the exponential growth of a single, excessively deep tree, it facilitates a far more efficient exploration for long-horizon tasks. This parallel, heuristic-guided merging is a new technique designed specifically to address the scaling limitations of diffusion planners.
>
> **3. The Preplan Composer: Amortized Inference via Reusable Plan Graphs**
>
> The Preplan Composer introduces a novel mechanism for optimizing inference-time efficiency by pre-computing a **compositional plan graph**.
>
> Crucially, this graph is designed to be **reusable across various tasks** within the same environment. This feature allows the significant computational overhead of planning to be **effectively amortized**, a substantial advantage for practical applications. To the best of our knowledge, this form of compositional graph reuse during the inference phase is a previously unexplored concept in the diffusion-based planning literature.
>
> **Concluding Remarks**
>
> These contributions, individually and collectively, represent a substantial advancement beyond applying existing techniques. We have developed a comprehensive framework that significantly enhances the applicability of diffusion planners to generalized inference-time scaling problems—particularly for tasks where the required plan length exceeds that seen during training. We are confident in the significance of this contribution and its potential to influence future work in long-horizon planning.
>
> > [W2]  The baselines are mostly weak replanning methods rather than SoTA long-horizon planners.
>
> We sincerely thank the reviewer for their constructive feedback on our experimental evaluation. We would like to take this opportunity to clarify a potential misunderstanding regarding the baselines used for comparison in our work.
>
> First, to address the concern about "weak replanning methods," **we went beyond simple replanning.** For established diffusion planners like Diffuser, Diffusion Forcing, and MCTD, we significantly strengthened them by integrating recent data-stitching techniques [1, 2]. These techniques are specifically designed to overcome the very training length limitations the reviewer highlighted. This ensures that our primary baselines are not naive implementations but are instead robust competitors that incorporate modern advancements.
>
> Second, regarding the lack of "SoTA long-horizon planners," our evaluation in **Table 1** explicitly includes comparisons against t**he most recent and relevant state-of-the-art methods.** We benchmarked C-MCTD against:
>
> - **Stitching Subtrajectory Diffuser (SSD):** A leading method that leverages a learned value function to address trajectory length limitations.
> - **Compositional Diffuser (CompDiffuser):** A SoTA planner that utilizes a specialized training paradigm for compositional planning.
> Our results, as detailed in Table 1, demonstrate that C-MCTD substantially outperforms *all* these methods—including the strengthened baselines and the current SoTA planners. For example, across all sizes (medium, large, giant) of the PointMaze and AntMaze environments, C-MCTD elevates performance dramatically, often turning success rates of 0-50% into 100%.
>
> This superior performance underscores the core contribution of our work: **C-MCTD achieves state-of-the-art results through a novel inference-time scaling stitching mechanism based on tree search, without requiring complex, specialized training schemes or auxiliary components like value functions.**
>
> > [W3] Results are inconsistent across environments - Distributed Composer sometimes outperforms Online Composer, sometimes doesn't, with no clear explanation.
>
> We thank the reviewer for their insightful comment. We would like to clarify the performance of our proposed methods.
>
> Our results demonstrate that the Distributed Composer **consistently outperforms** the Online Composer in the `pointmaze`and `antmaze` environments. This consistent improvement highlights its effectiveness in complex, sparse-reward settings.
>
> We acknowledge that the Distributed Composer showed a different performance trend in the Visual PointMaze environment. **As we briefly discussed in the manuscript (Page 9, Lines 340-342)**, we attribute this to the challenges of latent state misalignment that arise when stitching trajectories generated from diverse tree origins.
>
> > [Q1] Why not compare against recent methods that specifically address training length limitations, such as hierarchical planning approaches or other compositional methods?
>
> We thank the reviewer for their valuable feedback and the opportunity to clarify our comparisons with recent methods designed for long-horizon tasks with limited training data.
>
> **1. Comparison with Compositional Methods**
>
> We would like to gently point the reviewer to **Table 1** of our submission, where we conduct a detailed empirical comparison against two highly relevant compositional methods: SSD, a value-learning-based approach, and CompDiffuser, a state-of-the-art compositional planning method.
>
> - **Against SSD:** Our C-MCTD methods (Online, Distributed, and Preplan Composers) consistently and significantly outperform SSD across all tested environments (PointMaze and AntMaze).
> - **Against CompDiffuser:** All C-MCTD variants demonstrate comparable or superior performance in medium and large environments. Notably, in the most challenging 'giant' environments, our Preplan Composer surpasses CompDiffuser by a significant margin.
>
> We wish to emphasize a critical distinction that highlights the novelty of our work: CompDiffuser's performance relies on a diffusion planner trained with a highly specialized learning scheme. In contrast, a key contribution of C-MCTD is achieving these superior or comparable results using a **standard, pre-trained diffusion planner**. Our performance gains stem from our novel inference-time scaling and tree-search-based stitching framework, making C-MCTD a more general, modular, and practical approach.
>
> **2. Addressing Hierarchical Planning Methods**
>
> Regarding the comparison with hierarchical planning methods like the Hierarchical Multi-scale Diffuser (HMD) [2], a direct comparison was unfortunately not feasible as its source code has not been publicly released.
>
> However, to provide a principled and fair analysis, we isolated the core mechanism HMD employs to handle training length limitations: **dataset stitching**. We implemented this technique ourselves and applied it to strong diffusion planners (Diffuser, Diffusion Forcing, and MCTD) to create robust baselines. As shown in Table 1, our C-MCTD framework decisively outperforms these dataset-stitching-enhanced baselines. The performance gap is substantial, with C-MCTD achieving **over 50% higher success rates** in the challenging 'large' and 'giant' environments.
>
> This result demonstrates that the simple stitching of datasets is insufficient and that our proposed inference-time composition via tree search is a significantly more effective strategy for long-horizon planning. We hope this clarification underscores the strength and significance of our contributions.
>
> [1] Guanghe Li, Yixiang Shan, Zhengbang Zhu, Ting Long, and Weinan Zhang. Diffstitch: Boosting offline reinforcement learning with diffusion-based trajectory stitching. arXiv preprint arXiv:2402.02439, 2024.
>
> [2] Chen, Chang, et al. "Extendable long-horizon planning via hierarchical multiscale diffusion." arXiv preprint arXiv:2503.20102 (2025).

---

> > ### Author Response · Authors · 2025-08-06
> >
> > Dear Reviewer,
> >
> > We would like to reiterate our thanks for your insightful feedback. We have posted our author response, in which we sought to clarify the points you raised and address your concerns.
> >
> > With the discussion deadline approaching, it would be very helpful if you could confirm whether our rebuttal has sufficiently addressed the issues, or if any aspects require further clarification.
> >
> > Thank you for your valuable time and expertise.
> >
> > Sincerely,
> >
> > The Authors

---

### Official Review · Reviewer_ECBR · 2025-07-04

**Clarity:** 3
**Significance:** 3
**Originality:** 2
**Rating:** 4
**Confidence:** 3

**Summary:**

Building upon MCTD, the authors propose the C-MCTD framework, which incorporates three complementary strategies—Online, Distributed, and Preplan Composer—to perform compositional search over multiple short-horizon plans during the inference stage. This enables the generation of long-horizon trajectories exceeding 10× the training horizon. Experiments across OGBench’s Point-/Ant-Maze, vision-based mazes, and robotic block manipulation tasks demonstrate significant performance improvements over baselines such as MCTD-Replan and CompDiffuser.

**Questions:**

- the paper only reports inference-time runtime (Table 2), how long does the offline graph construction take?
- the author states that the Online Composer relies on mechanisms such as guidance sets and fast replanning, but their specific contributions to performance are not demonstrated.

**Ethical Concerns:**

["NO or VERY MINOR ethics concerns only"]

**Limitations:**

yes (section 5.4)

**Quality:**

3

**Strengths And Weaknesses:**

Strength:
The paper extends the MCTD to composable inference-time reasoning, allowing long-horizon plan generation without retraining.

Weakness:
The paper lacks ablation studies on key design components of the Composers—for instance, the Fast Replanning and Plan-Cache modules in the Online Composer.

---

> ### Author Rebuttal · Authors · 2025-07-31
>
> We appreciate the reviewer's time and effort in reviewing our manuscript.
>
> > [W1] The paper lacks ablation studies on key design components of the Composers—for instance, the Fast Replanning and Plan-Cache modules in the Online Composer.
>
> We thank the reviewer for their constructive feedback. We appreciate the opportunity to clarify our analysis of the key components of the Online Composer and provide further validation.
>
> **1. Ablation Study on the Plan-Cache Module**
>
> We would like to clarify that an ablation study for the Plan-Cache module is provided in **Appendix B.1**. These experiments were conducted on the Robot Arm Cube Manipulation tasks. As detailed in Table 5, our findings show that while the benefit of the Plan-Cache is modest for relatively simple tasks (e.g., single or double object manipulation), its utility increases exponentially with task complexity. Notably, in the most complex quadruple cubes manipulation task, the Plan-Cache **achieves a ~6x reduction in runtime** (from 1432.8s to 242.8s), demonstrating its critical role in scaling our method to challenging, long-horizon problems.
>
> **2. Ablation Study on the Fast Replanning Module**
>
> Following the reviewer's valuable suggestion, we have conducted a new ablation study on the Fast Replanning (FR) module. We evaluated the performance of the Online Composer with and without Fast Replanning across various search budgets in our maze environments. The numbers in parentheses indicate the maximum search iterations allowed. We did not report results on PointMaze Giant as the performance difference was negligible.
>
> | Environment | W/ FR (200) | W/o FR (200) | W/ FR (100) | W/o FR (100) | W/ FR (50) | W/o FR (50) |
> | --- | --- | --- | --- | --- | --- | --- |
> | PointMaze Medium | **96±8%** | 82±6% | **94±9%** | 76±12% | **92±10%** | 63±15% |
> | PointMaze Large | **84±15%** | 54±13% | **82±14%** | 49±10% | **82±19%** | 32±11% |
>
> The results unequivocally demonstrate the importance of Fast Replanning.
>
> - **Consistent Improvement:** Across all settings, enabling Fast Replanning leads to a significant performance improvement.
> - **Efficiency under Constraint:** The performance gap widens dramatically as the search budget shrinks. For instance, in PointMaze Large with a budget of 50, our method with Fast Replanning maintains a high success rate of 82%, whereas the performance of the version without it collapses to 32%.
>
> This analysis highlights that Fast Replanning is not just a minor optimization but a crucial component that makes our tree search algorithm robust and sample-efficient, by enabling effective value propagation throughout the search tree.
>
> We will add this new ablation study to the appendix in the revised version. We thank the reviewer again for their insightful comments, which have helped us further strengthen our paper.
>
> > [Q1] the paper only reports inference-time runtime (Table 2), how long does the offline graph construction take?
>
> We thank the reviewer for this insightful question regarding the offline graph construction time. We omitted this from the main paper as it is a one-time, pre-inference step, but we agree that providing this information adds valuable context. We will include these details in the final version of the paper.
>
> We measured the runtime for building the plan graph for each PointMaze environment on a system with 8 NVIDIA GeForce RTX 4090 GPUs. The results are as follows:
>
> - **PointMaze Medium (10 nodes):** 46.16 seconds
> - **PointMaze Large (30 nodes):** 83.18 seconds
> - **PointMaze Giant (70 nodes):** 72.69 seconds
>
> Notably, the construction for the 'Giant' environment is faster than for 'Large'. This is because the higher density of nodes (70 vs. 30) allows us to set a shorter maximum length for the plans connecting each pair of nodes, which in turn reduces the per-pair search complexity.
>
> The key takeaway is that this offline cost represents **a highly practical one-time investment**. For context, in the 'Large' environment, **the entire graph construction (83.18s) takes less time than a single planning instance with our Online Composer (91.2s, as reported in Table 2).**
>
> Once this graph is constructed for a given environment, it is reused for all subsequent planning tasks, including any unseen start-goal pairs. **This one-time computational cost is therefore effectively amortized across numerous runs**, leading to the significant inference-time efficiency gains of our Preplan Composer. This strategic trade-off—a modest, one-time offline computation for a substantial and repeated online speedup while maintaining high performance—is a central achievement of our proposed method.
>
> > [Q2] the author states that the Online Composer relies on mechanisms such as guidance sets and fast replanning, but their specific contributions to performance are not demonstrated.
>
> We thank the reviewer for their insightful comment. To clarify the distinct contributions of our proposed mechanisms, **we have already included an ablation study on Fast Replanning in the above comment for [W1]**, and we now present a new ablation study on the guidance set.
>
> In our framework, C-MCTD provides the planner with a guidance set—e.g., [0, 0.1, 0.5, 1, 2] for the PointMaze tasks—as a meta-action. This allows for effective integration with inference-time scaling planners. To experimentally isolate the contribution of this component, we compared our default approach (providing the full set) against providing the planner with a single, fixed guidance level.
>
> The results are summarized in the table below:
>
> | Environment | Default (Guidance Set) | Fixed: 0.1 | Fixed: 0.5 | Fixed: 1 | Fixed: 2 |
> | --- | --- | --- | --- | --- | --- |
> | PointMaze Medium | **93±9%** | 80±0% | 90±10% | **98±6%** | 83±8% |
> | PointMaze Large | **82±17%** | **77±14%** | 52±13% | 27±10% | 20±0% |
>
> The results reveal a clear trade-off. For the `PointMaze-Medium` environment, which requires moderately complex planning, higher fixed guidance levels (0.5 or 1.0) yield strong performance, comparable to using the full set. In contrast, for the more challenging `PointMaze-Large` environment, a low fixed guidance level (0.1) is required to achieve a competitive score.
>
> Crucially, our default method of providing a guidance set achieves robustly high performance across both environments. This demonstrates that **the guidance set is a vital mechanism that empowers the planner to adaptively select the most suitable guidance strength for a given task.** It is this adaptability that ensures effective and versatile integration with inference-time planners across varying levels of environmental complexity.
>
> We will incorporate this analysis and the table into the revised manuscript to make the contribution of the guidance set explicit.

---

> > ### Author Response · Authors · 2025-08-06
> >
> > Dear Reviewer,
> >
> > Thank you again for your valuable and constructive review. We have submitted our rebuttal and hope that it has addressed your concerns.
> >
> > As the discussion period is ending soon, we would be very grateful if you could let us know whether our response is satisfactory or if there are any remaining points you would like to discuss.
> >
> > Thank you for your time and consideration.
> >
> > Sincerely,
> >
> > The Authors

---

### Note · Authors · 2025-08-16

Dear Area Chair and Reviewers,

We sincerely thank the reviewers for their constructive feedback, which has significantly improved our manuscript, and the Area Chair for this final response opportunity.

We are pleased that our productive discussions successfully addressed the concerns of three reviewers, **as reflected in their positive follow-up comments and score updates**. We commit to incorporating their suggestions into the revised version, such as enhanced experimental details and visualizations (Reviewer **Q2ax**), comprehensive runtime comparison against all baselines and a clearer description of the Preplan Composer's methodology (Reviewer **y79K**), and a clearer problem formulation with analysis of online planning in stochastic environments and underlying assumptions (Reviewer **rpAX**).

Unfortunately, we could not have a follow-up discussion with Reviewers **ECBR** and **NuJi**. However, we wish to confirm that our rebuttal addressed their primary concerns:

- **For Reviewer ECBR**: Our rebuttal provided detailed ablation studies for our key design components (Fast Replanning, Plan-Cache, guidance set) to clarify their performance contributions. We also analyzed the offline graph construction time in relation to the inference-time runtime (Table 2).

- **For Reviewer NuJi**: We clarified our comparisons against very recent state-of-the-art methods [1-3]. On the topic of novelty, we re-emphasized that our work is not a simple application of existing techniques. It introduces significant improvements—such as tree stitching and a reusable plan graph for amortized inference—to tackle long-horizon generation, a key challenge unaddressed by prior diffusion-based planners (e.g., MCTD).

We are confident that the review process has made our paper stronger and more impactful. We firmly believe our work represents a significant step forward in enabling high-performing, diffusion-based planning far beyond the training horizon.

Thank you again for your time and positive consideration of our work.

Sincerely,

The Authors

[1] Stitching sub-trajectories with conditional diffusion model for goal-conditioned offline rl, AAAI 2024.

[2] Generative trajectory stitching through diffusion composition, ArXiv 2503.05153.

[3] Extendable long-horizon planning via hierarchical multiscale diffusion, ArXiv 2503.20102.

---

### Decision · Program_Chairs · 2025-09-17

**Decision:**

Accept (spotlight)

**Comment:**

This paper proposes Compositional Monte Carlo Tree Diffusion (C-MCTD), an extension of MCTD designed to overcome limited planning horizons by stitching together multiple short-horizon plans. The framework introduces three complementary strategies—Online Composer (sequential hierarchical stitching), Distributed Composer (parallel plan trees with connections), and Preplan Composer (prebuilt plan graphs)—to enable compositional search during inference. This allows the generation of trajectories more than 10× longer than the training horizon. Experiments on maze navigation, vision-based planning, and robotic manipulation tasks demonstrate that C-MCTD significantly outperforms replanning baselines such as MCTD-Replan and CompDiffuser in both performance and efficiency.

The main strength of the paper, which is acknowledge by the reviewers, is that an important limitation of MCTD, the limited planning horizon, is overcome. The concerns are mostly in terms of presentation and also some aspects in the evaluation.

There has been an extensive discussion with the reviewers, and mostly the opinion is that this is a paper worthy of being accepted. I will follow this recommendation. However, it seems that the paper could benefit from some more clarifications and discussions, which is why I am hesitating to recommend for an oral presentation. The authors should address the reviewers' concerns for the CR version and their presentation.